# Revealing the three-dimensional arrangement of polar topology in nanoparticles

Chaehwa Jeong [1], Juhyeok Lee [1,2,3], Hyesung Jo [1], Jaewhan Oh[1], Hionsuck Baik[4], Kyoung-June Go [5], Junwoo Son [6], Si-Young Choi [5,7], Sergey Prosandeev[8], Laurent Bellaiche [8] & Yongsoo Yang [1,9] ✉

In the early 2000s, low dimensional ferroelectric systems were predicted to have topologically nontrivial polar structures, such as vortices or skyrmions, depending on mechanical or electrical boundary conditions. A few variants of these structures have been experimentally observed in thin film model systems, where they are engineered by balancing electrostatic charge and elastic distortion energies. However, the measurement and classification of topological textures for general ferroelectric nanostructures have remained elusive, as it requires mapping the local polarization at the atomic scale in three dimensions. Here we unveil topological polar structures in ferroelectric $BaTiO_3$ nanoparticles via atomic electron tomography, which enables us to reconstruct the full three-dimensional arrangement of cation atoms at an individual atom level. Our three-dimensional polarization maps reveal clear topological orderings, along with evidence of size-dependent topological transitions from a single vortex structure to multiple vortices, consistent with theoretical predictions. The discovery of the predicted topological polar ordering in nanoscale ferroelectrics, independent of epitaxial strain, widens the research perspective and offers potential for practical applications utilizing contact-free switchable toroidal moments.

Ferroelectric crystals have the ability to exhibit spontaneous polarizations that can be switched by external electric fields. A range of physical properties is related to ferroelectricity, including piezoelectricity, pyroelectricity, flexoelectricity, and dielectricity, which make ferroelectrics useful for various electronic devices, such as memories, sensors, and capacitors[1–7]. Typically, bulk ferroelectric crystals exist in a paraelectric state at high temperatures, but as the temperature decreases below the Curie temperature, they make a transition into ferroelectric states. The ferroic states are often accompanied by multiple domains with varying spontaneous polarizations, and the properties of ferroelectrics are significantly affected by the internal domain structures.

[1]Department of Physics, Korea Advanced Institute of Science and Technology (KAIST), Daejeon 34141, Republic of Korea. [2]Energy Geosciences Division, Lawrence Berkeley National Laboratory, Berkeley, CA 94720, USA. [3]National Center for Electron Microscopy, Molecular Foundry, Lawrence Berkeley National Laboratory, Berkeley, CA 94720, USA. [4]Korea Basic Science Institute (KBSI), Seoul 02841, Republic of Korea. [5]Department of Materials Science and Engineering, Pohang University of Science and Technology (POSTECH), Pohang 37673, Republic of Korea. [6]Department of Materials Science and Engineering, Research Institute of Advanced Materials, Seoul National University Seoul 08826, Republic of Korea. [7]Center for Van der Waals Quantum Solids, Institute for Basic Science (IBS), Pohang 37673, Republic of Korea. [8]Smart Ferroic Materials Center (SFMC), Physics Department and Institute for Nanoscience and Engineering, University of Arkansas, Fayetteville, AR 72701, USA. [9]Graduate School of Semiconductor Technology, School of Electrical Engineering, Korea Advanced Institute of Science and Technology (KAIST), Daejeon 34141, Republic of Korea. ✉e-mail: yongsoo.yang@kaist.ac.kr

Ferroelectric nanostructures differ from their bulk counterparts; uncompensated charges at their surfaces create a depolarizing field, which can be strong enough to suppress spontaneous polarization and prevent ferroelectric nanostructures from exhibiting a ferroelectric state even at low temperatures[8–11]. To induce polarizations, external electric fields or conducting boundary conditions are required to screen the depolarizing field[12,13]. Therefore, the occurrence and behavior of ferroelectric ordering in low-dimensional systems have been a topic of interest.

With the development of effective Hamiltonian and phase-field methods, the existence of vortices[14,15] or other topological structures such as flux-closures[16], bubbles[17], skyrmions[18], merons[19], polar waves[20], and topological defects[21] was predicted from simulations for different types of nanostructures including films and nanodots. Many experimental efforts were also devoted to fabricating various ferroelectric nanostructures and observing polar topologies, and the use of advanced transmission electron microscopy (TEM) and piezoresponse force microscopy techniques has enabled the observation of a periodic array of full flux-closure quadrants in $PbTiO_3/SrTiO_3$ multilayer/superlattice systems[22], as well as other predicted structures[23–25].

However, these techniques are often restricted to surfaces or lower dimensional projections, or resolutions at a non-atomic scale[26], and are inadequate for probing three-dimensional (3D) polarization structures within the ferroelectric nanostructures. Due to this limitation, most of the polar topology structures have been determined from cross-sections of thin films for which the epitaxial strain plays a crucial role. Although the existence of polar vortices in ferroelectric nanostructures, including nanodots, nanowires, and nanodisks, has been predicted without the need for epitaxial strain[14,15,27], experimental observation of such ordering and their size-dependent structural transitions has not yet been achieved. Furthermore, a complete classification of 3D topological textures would require a true 3D mapping of local polarizations, because predicting higher dimensional structures from lower dimensional projections can often be misleading[28,29]. Therefore, it is necessary to directly image the 3D polarization structures to fully understand the polar topology.

Typically, ferroelectric polarizations are directly related to local atomic displacements within the unit cell (inversion symmetry breaking). If full 3D atomic coordinates of all the atoms within ferroelectric nanostructures can be determined at high precision, atomic scale 3D local polarization can be fully mapped. Here we report the experimentally observed topological ferroelectric ordering within $BaTiO_3$ nanoparticles of approximately 8.8 nm and 10.1 nm diameters at an 3D individual atom level, which was achieved via precise determination of the 3D coordinates of cation atoms using atomic electron tomography (AET)[28–31].

## Results

### Determination of 3D atomic structures of BaTiO₃ nanoparticles

The $BaTiO_3$ nanoparticles were fabricated using hydrothermal synthesis. Their structure and composition were confirmed using energy dispersive X-ray spectroscopy (EDS) mapping and powder X-ray diffraction (PXRD) (see Supplementary Fig. 1). After drop-casting the particles onto a SiN membrane, an amorphous carbon layer of approximately 2.0 nm thickness was deposited, followed by a vacuum cleaning of 10 h at 150 °C (Methods). Two nanoparticles, with average diameters of ~8.8 nm (Particle 1) and ~10.1 nm (Particle 2), were selected for annular dark-field scanning transmission electron microscopy (ADF-STEM) tilt-series measurements to conduct AET (see Supplementary Fig. 2 and Methods). The GENFIRE algorithm[32] was used to reconstruct the atomic resolution 3D tomograms of both nanoparticles (Methods). Note that the total electron dose for ADF-STEM data acquisition was optimized to minimize the beam damage (see Supplementary Fig. 3) and the contrast from oxygen atoms was too weak to be detected under the given dose condition.

The 3D atomic coordinates and chemical species of Ba and Ti atoms in each nanoparticle were determined using atom tracing and classification techniques, with a precision of 38.3 pm for Particle 1 and 34.4 pm for Particle 2, as determined via multislice simulations (Methods). The reliability of the structures was supported by the reasonable level of R-factors, which were calculated by comparing the measured tilt-series data with the forward-projections from the determined atomic structures (Methods).

The resulting 3D atomic models of Particle 1 and Particle 2, depicted in Fig. 1a and Supplementary Fig. 4a, revealed the alternating Ba and Ti atomic layers along the <001> directions. Figure 1b–d and Supplementary Fig. 4b–d further show the 1.07 Å thick internal slices of the 3D tomogram along the [001] (Fig. 1b–d) or [010] (Supplementary Fig. 4b–d) directions for three consecutive atomic layers near the core of the particles. The intensity distribution of the tomogram slices and traced atom positions clearly represent the atomic planes of square lattices, consistent with expected $ABO_3$ perovskite cation ordering. Over 99.5% of the 3D cation atomic coordinates obtained could be successfully fitted into a body-centered cubic (bcc) lattice (Methods), which is consistent with the expected cubic phase of $BaTiO_3$ with sizes smaller than 30 nm[33].

Before the 3D displacement analysis, we first conducted the two-dimensional (2D) Ti displacement mapping of linearly projected 3D volumes (along [100], [010], and [001]) generated from the experimentally determined 3D atomic structures (Fig. 1e–g and Supplementary Fig. 4e–g). The grayscale background indicates the projected intensity, where Ba and Ti columns can be distinguished from the relative difference in contrasts (Ti columns are weaker). To quantify the local Ti displacement directions, we calculated the deviation of the identified Ti column positions from the geometric centers of the four neighboring Ba column positions[34]. Note that the projections along the [100] direction exhibit minor indications of vortex-shaped ordering (Fig. 1e and Supplementary Fig. 4e), but no such ordering can be found from the projections along the other directions (Fig. 1f, g and Supplementary Fig. 4f, g). We also conducted several 2D-ADF-STEM measurements for other $BaTiO_3$ particles along <001> zone axes, and they also showed no vortex-like features (Supplementary Fig. 5), consistent with previous 2D-projection-based studies[35,36].

### 3D local polarization maps of the nanoparticles

Figure 2 and Supplementary Fig. 6 present sliced maps showing the 3D distribution of Ti atomic displacements for the Particle 1 (8.8 nm $BaTiO_3$). In Fig. 2a, we show the in-plane atomic displacement direction maps of representative Ti atomic layers sliced along the [100] direction. The displacements were calculated as the deviation of each 3D Ti position from the geometric center of the eight neighboring Ba atomic positions and subsequently averaged using a Gaussian kernel (Methods). Note that the application of kernel-based local averaging effectively improves precision, reducing the error in the locally-averaged displacement field to about 4.9 pm for Particle 1 and 4.3 pm for Particle 2. This level of precision is adequate to describe the global polar ordering behavior, considering the typical ferroelectric local displacement level (tens of picometers). However, as a trade-off, it results in a slightly lower 3D spatial resolution of approximately 0.9 nm[31]. The polarization maps before kernel averaging can also be found in Supplementary Fig. 6.

The in-plane displacement direction maps clearly reveal that unidirectional polar order dominates in the top half of the particle while a counterclockwise vortex exists in the bottom half. This corresponds to the theoretically predicted polar-toroidal multiorder (PTMO) state, in which the polar and toroidal orders coexist[37]. In real samples, due to the presence of various screening sources such as charge carriers arising from oxygen vacancies, the electrostatic boundary condition can exhibit a partially screened state, rather than being perfectly short-circuited or fully open-circuited. In this case, the

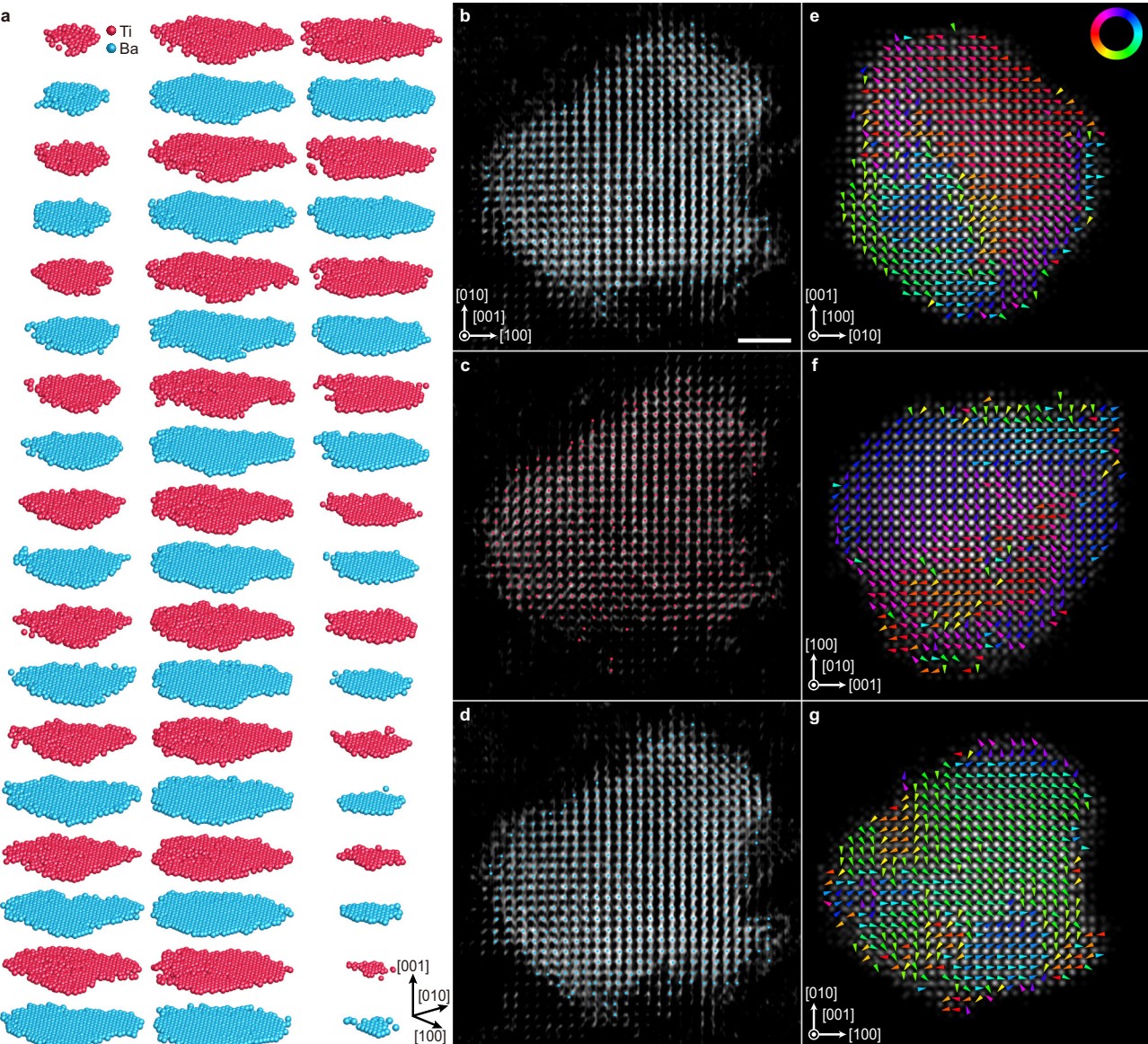

**Fig. 1 | Determination of the 3D atomic structure of a 10.1 nm BaTiO₃ nanoparticle (Particle 2). a** Cation atomic layers along the [001] crystallographic direction, showing the 3D atomic arrangements of alternating Ba and Ti layers. **b–d** 1.07 Å thick internal slices of the 3D tomogram along the [001] direction for three consecutive atomic layers near the core of the particle. The intensity of the sliced tomogram is depicted in grayscale, while the atomic coordinates of Ba and Ti are represented with blue and red dots, respectively. Scale bar, 2 nm. **e–g** Linear projections of a 3D intensity volume representing the determined 3D atomic structure, projected along [100] (**e**), [010] (**f**), and [001] (**g**) directions, respectively. The grayscale background indicates the projected intensity, where Ba and Ti columns can be distinguished by their relative intensities (Ti columns show weaker contrasts). The arrows represent the Ti displacements, calculated as the deviation of the identified Ti column positions from the geometric centers of the four neighboring Ba column positions. The arrows are colored based on the displacement direction, as given in the color wheel in (**e**).

vortex core can be off-centered rather than being at the center of the particle, resulting in toroidal order near the vortex center and polar ordering away from it. Note that no clear vortex structures can be found if the particle is sliced along [010] or [001] directions (Supplementary Fig. 6), indicating that the vortex line is predominantly oriented along the [100] direction.

In Fig. 2b, the magnitudes of the kernel-averaged 3D Ti atomic displacements and calculated local polarization values are plotted (Methods). The range of 3D Ti atomic displacements (and the resulting polarization values) is consistent with the previously observed values in the nanosized BaTiO₃[38,39]. The region near the core of the vortex (indicated by blue arrows) shows relatively smaller local polarizations compared to other regions where the vortex does not appear, which is also in agreement with the result from phase-field simulations[27].

Based on the 3D displacement vectors of all Ti atoms, we calculated the average displacement vector, which was found to be 18.7 ± 0.50 pm in magnitude along the direction of (0.27, −0.96, 0.07), resulting in the estimated net polarization per unit cell of 0.40 ± 0.13 C m⁻² (Methods). This value is slightly larger than that of bulk BaTiO₃[40] (0.25 C m⁻²). We also calculated the overall magnitude of the displacement vector by fitting our distribution to a generalized extreme value distribution[41], considering the asymmetry of the measured Ti atomic displacements (Supplementary Fig. 7), which was determined to be 43.5 pm. Several experimental and simulation studies have indicated that the magnitude of Ba-Ti displacements increases as the size of the system decreases up to a certain point (beyond which the ferroelectric ordering can be suppressed)[36,38,42]. For instance, a mean Ba-Ti displacement of -18.9 pm was observed from a

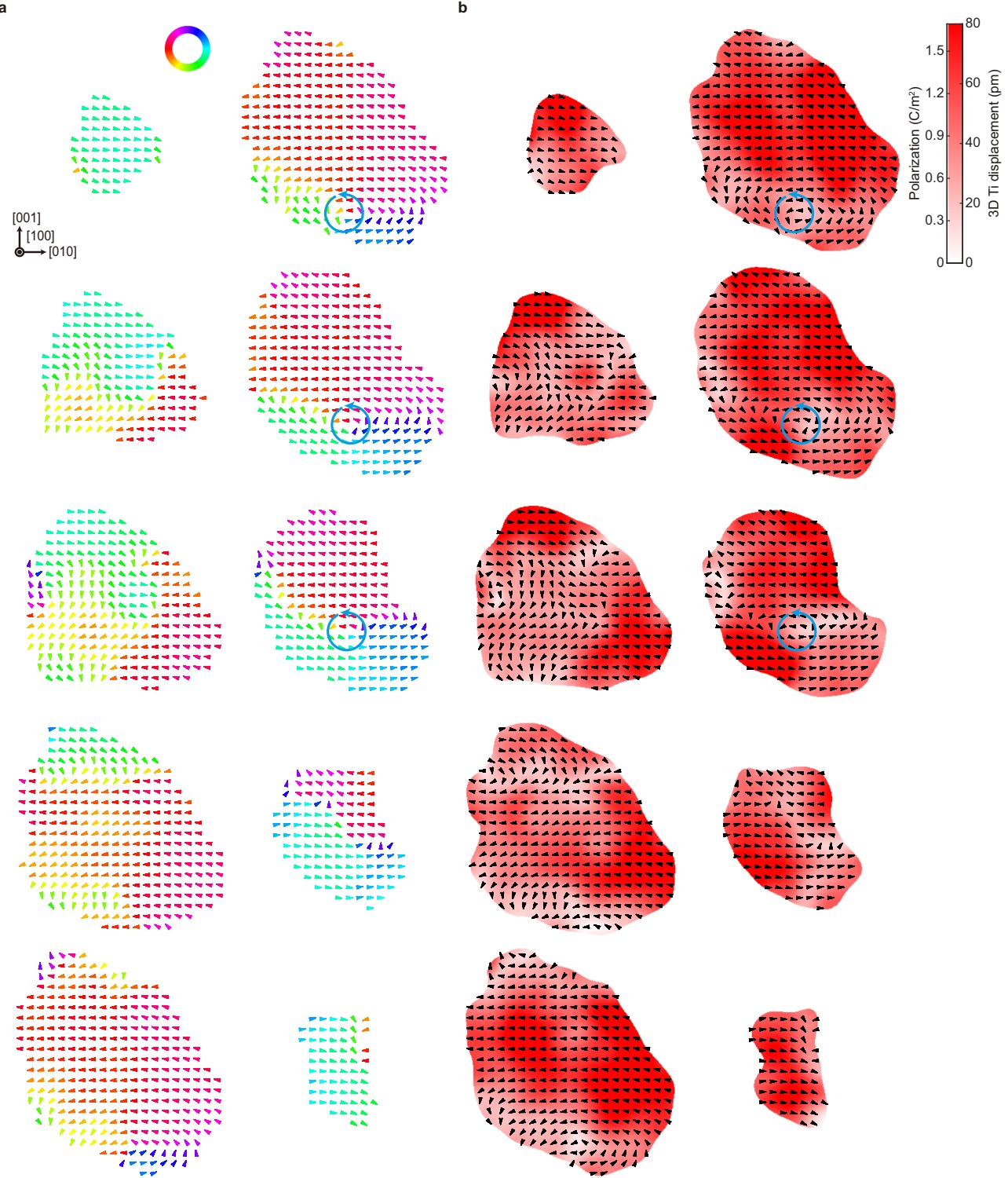

**Fig. 2 | Sliced maps showing the 3D distribution of Ti atomic displacements for the 8.8 nm BaTiO₃ (Particle 1). a** In-plane atomic displacement direction maps of the representative Ti atomic layers sliced along the [100] direction. Note that the spacing between the plotted layers is 2 unit cells. The displacements were calculated as the deviation of each 3D Ti position from the geometric center of the eight neighboring Ba atomic positions and subsequently averaged using a Gaussian kernel (Methods), and their in-plane directions are plotted (the color of arrows also indicates the direction as given in the color wheel). The existence of a single counterclockwise vortex can be clearly seen (marked with blue arrows). Scale bar, 2 nm. **b** Maps showing the magnitude of the kernel-averaged 3D Ti atomic displacements and local polarization for the corresponding Ti atomic layers in (**a**). The local polarizations were obtained through a linear relation with 3D Ti atomic displacements. The in-plane directions of the displacements are overlaid as black arrows, and the positions of counterclockwise vortices given in (**a**) are consistently marked with blue arrows.

26 nm BaTiO$_3$ nanoparticle[36], and ~30 pm was found from a 4 × 4 nm$^2$ BaTiO$_3$ nanocluster[38]. Considering that these experimental observations are for 2D projected displacements, our measured 3D values are consistent with these previous studies. Having the full 3D local polarization distribution further enables us to calculate the net toroidal moment vector per unit cell **G** (Methods). The magnitude and direction of the vector were 0.22 ± 0.07 $e$ Å$^{-1}$ and (0.89, −0.40, 0.20), respectively. Here, the direction of the toroidal moment is not fully aligned with the [100] direction. This is also responsible for the PTMO state mentioned above, resulting in the shift of the vortex center positions in different 2D slices (eventually hitting the surface of the particles), as shown in Fig. 2.

Similar analyses were conducted for the Particle 2 (10.1 nm), as described in Fig. 3 and Supplementary Fig. 8. Notably, the in-plane displacement direction maps along the [100] direction of this particle show multiple vortices, as can be seen in several consecutive layers near the center of the nanoparticle (Fig. 3a). This behavior is precisely what is expected for ferroelectric nanoparticles, transitioning from a single vortex structure to multiple vortex structures as the particle size increases[27]. The magnitudes of 3D Ti atomic displacements and local polarization values also exhibit reasonable values, with expected suppression near the core of the vortices (Fig. 3b).

The average displacement vector for the Particle 2 was found to be 29.0 ± 0.34 pm in magnitude along the direction of (0.61, −0.78, 0.10), and the net polarization per unit cell was estimated to be 0.61 ± 0.20 C m$^{-2}$. The overall magnitude of the displacement vector for Particle 2 was determined to be 40.5 pm (Supplementary Fig. 7), slightly smaller than that for Particle 1, consistent with previous studies of the size effect on the magnitude of the displacement vector[42]. For Particle 2, the magnitude and direction of the net toroidal moment vector per unit cell **G** were 0.51 ± 0.16 $e$ Å$^{-1}$ and (0.88, −0.46, −0.04), respectively. While the direction is more or less consistent with that of Particle 1, the magnitude is about twice as large as that of Particle 1. However, the distribution of the magnitude of local toroidal moments is similar for both particles (Supplementary Fig. 9a, b). Therefore, the difference in the net moments originates from the fact that the local toroidal moments of Particle 2 show a stronger tendency to align parallel to the direction of the net moment (Supplementary Fig. 9c, d). Moreover, the contribution of the surface atoms to the net moments is higher compared to that of the core atoms for both particles (Supplementary Fig. 9e, f), resulting in a larger net moment for the larger particle, which has a greater number of surface atoms.

Since the nanoparticles show substantial cation displacement and consequent local polarization which typically accompany local tetragonal distortion, we further performed local structural analysis regarding the local tetragonality (Methods). Although no distinct long-range tetragonal distortion can be observed from the PXRD analysis (see Supplementary Fig. 1g and Methods), there are localized areas with high tetragonality that cannot arise from a cubic phase (Supplementary Figs. 10 and 11). Our findings, where the nanosized BaTiO$_3$ exhibits a cubic-like feature when ensemble averaged but shows substantial distortion on the short-range scale, are consistent with the results in previous studies[43]. Additionally, a comparison of the tetragonality map (Supplementary Fig. 10) with the 3D displacement magnitude map (Figs. 2 and 3) reveals that areas with low tetragonality (i.e., $c/a$ ratio being close to 1) can also exhibit large Ti atomic displacement and resulting polarization. In nanostructured ferroelectric systems, unlike in bulk materials, an unusually strong atomic displacement has been observed in previous studies, and it does not always correlate with the tetragonality or $c$-axis orientations, consistent with our findings[44–46].

### Effective Hamiltonian simulations

To corroborate the experimental findings, we conducted effective Hamiltonian simulations for similarly sized nanoparticles (Particle $\widetilde{1}$ and Particle $\widetilde{2}$; see Methods). For the Particle $\widetilde{1}$ whose size is comparable to that of Particle 1, the toroidal moment is predicted to be 0.16 $e$ Å$^{-1}$ in magnitude along (−0.81, 0.44, −0.39), which agrees reasonably well with the experiment in terms of both magnitude and direction (sign of each component is not important here because there are eight degenerate solutions of equal free energy arising from the + or − sign). For Particle $\widetilde{2}$ (similar in size to Particle 2), the calculations give the toroidal moment of magnitude 0.18 $e$ Å$^{-1}$ (about a third of the experimental value) along the direction of (0.86, 0.44, −0.25).

Note that the toroidal moment, in both simulations, depends on temperature and grows in amplitude, when cooling the systems down (Supplementary Fig. 12), eventually reaching (−0.20, 0.20, −0.19) $e$ Å$^{-1}$ and (0.21, 0.22, −0.22) $e$ Å$^{-1}$ for Particle $\widetilde{1}$ and Particle $\widetilde{2}$, respectively, at 5 K.

This confirms that the larger particle exhibits a greater magnitude of toroidal moment than the smaller one, as observed in the experiments, although the difference is somewhat less pronounced. The discrepancies between the experiment and simulation can be attributed to the fact that the calculations were performed for ideal nanoparticles without considering surface reconstruction, defects, and surface adatoms (carbon atoms can easily contaminate the surface during the experiment, which cannot be detected in our ADF-STEM images). Moreover, the depolarizing field results in the absence of polarization in our simulations, contrasting with this and other experiments[35,36,47] which show the presence of polarization. This is not surprising because there can be an effective screening of the depolarizing field in the experimental system, which may result from charge carriers induced by the presence of oxygen vacancies[48–50], partially screened electrostatic boundary condition, or/and lateral strain, leading to the aforementioned PTMO states. Such a comparison tells us that the performed experiment brings a new challenge for the future theoretical investigations.

We also found a single vortex structure in the Particle $\widetilde{1}$ and multiple vortices in the Particle $\widetilde{2}$ from the 2D slices perpendicular to the [100] direction, as illustrated in Fig. 4, showing an excellent agreement with the experimental findings (Figs. 2 and 3). These vortices arise from a connection of the domains having different directions of the local polarization. For example, the vortex in the Particle $\widetilde{1}$ connects 4 domains having in-plane dipoles along $\hat{\mathbf{y}} + \hat{\mathbf{z}}$, $\hat{\mathbf{y}} − \hat{\mathbf{z}}$, $−\hat{\mathbf{y}} − \hat{\mathbf{z}}$, and $−\hat{\mathbf{y}} + \hat{\mathbf{z}}$, where $\hat{\mathbf{y}}$ and $\hat{\mathbf{z}}$ are unit vectors along the $y$- and $z$-axis, respectively. Moreover, the presence of more vortices in the Particle $\widetilde{2}$ tells us that the domain structures observed in these experiments and simulations are more complex than the texture resulting from a simple elongation of dots[51,52].

## Discussion

In this work, we have determined the full 3D atomic coordinates of cation atoms for ferroelectric BaTiO$_3$ nanoparticles of two different sizes via AET. This has unveiled the theoretically predicted toroidal ordering of local polarizations in the absence of epitaxial strain, together with a hint of the size-dependent phase transition in terms of the number of vortices. Our approach provides a full 3D polarization distribution for ferroelectric nanostructures, allowing for many more analyses to be conducted on various nanosized ferroelectric systems. For instance, topological polar structures and their transitions can be experimentally measured and classified by tuning the size, shape, and surface boundary conditions of nanostructures, leading to a better understanding and control over topological orderings. Additionally, a curl of in-plane polarization (the curl of the normalized in-plane Ti displacement fields for each atomic plane; see Methods) can be calculated to qualitatively visualize the tendency of the local rotations of polarization vectors in the 2D slices through the particles. A few representative maps are shown in Supplementary Fig. 13, where local curl vectors exhibit large magnitudes near the vortex cores of both particles. Interestingly, the curl vectors in Particle 2 alternately flip the

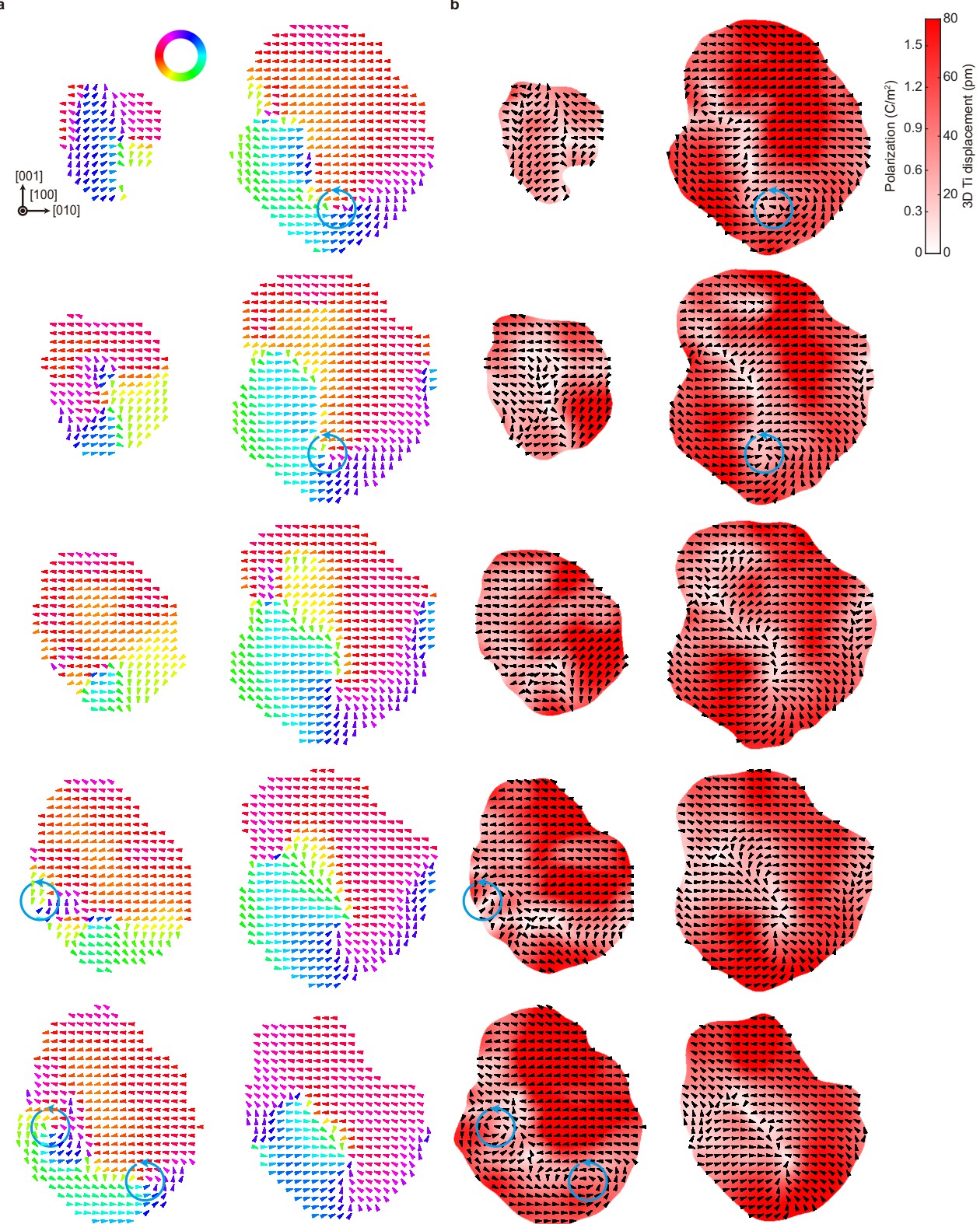

**Fig. 3 | Sliced maps showing the 3D distribution of Ti atomic displacements for the 10.1 nm BaTiO₃ (Particle 2). a** In-plane atomic displacement direction maps of the representative Ti atomic layers sliced along the [100] direction. Note that the spacing between the plotted layers is 2 unit cells. The displacements were calculated as the deviation of each 3D Ti position from the geometric center of the eight neighboring Ba atomic positions and subsequently averaged using a Gaussian kernel (Methods), and their in-plane directions are plotted (the color of arrows also indicates the direction as given in the color wheel). The existence of multiple counterclockwise vortices can be clearly seen (marked with blue circular arrows). Scale bar, 2 nm. **b** Maps showing the magnitude of the kernel-averaged 3D Ti atomic displacements and local polarization for the corresponding Ti atomic layers in (**a**). The local polarizations were obtained through a linear relation with 3D Ti atomic displacements. The in-plane directions of the displacements are overlaid as black arrows, and the positions of counterclockwise vortices given in (**a**) are consistently marked with blue arrows.

**a**

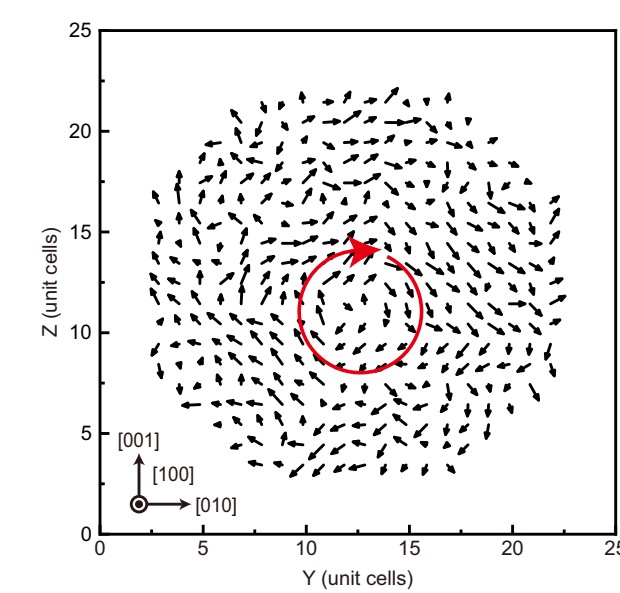

**b**

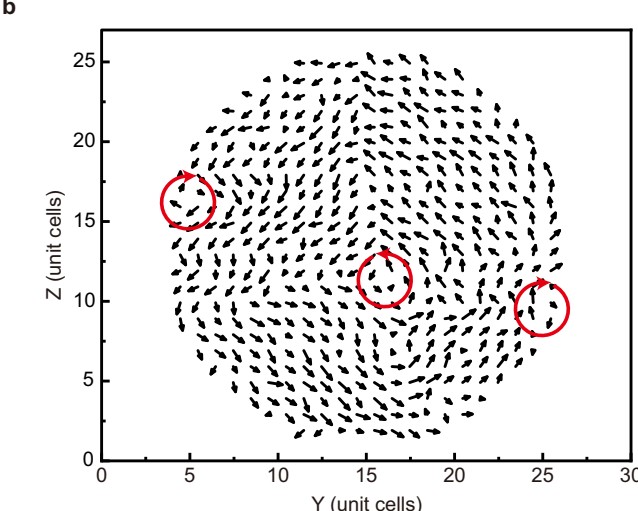

**Fig. 4 | Simulated local dipole patterns in BaTiO₃ nanoparticles at 300 K.**
**a, b** Simulated local dipolar structure maps at 300 K, representing the slices perpendicular to the [100] direction for the smaller (Particle $\tilde{1}$, 8.8 nm) (**a**), and larger (Particle $\tilde{2}$, 9.7 nm) (**b**) BaTiO₃ nanoparticles. The existence of a single vortex and multiple vortices (marked with red circular arrows) can be clearly observed from the smaller and larger nanoparticles, respectively, consistent with the experiments. The lattice constant of 5-atom unit cell was 4.01 Å in the simulation.

direction from $+x$ to $-x$ multiple times around a small region, and the polarization ordering pattern near this region certainly resembles that of an antivortex (Supplementary Fig. 13). In addition to the vortices and antivortices, convergent hedgehog-type structures can also be found in some regions of Particle 2 (Supplementary Fig. 13). The difference in the number of vortices (one vortex for Particle 1 and two vortices for Particle 2) originates from the size-dependent topological transitions from single vortex-like structures to those of multiple vortices, as predicted from effective Hamiltonian and phase-field simulations[15,27,53]. In particular, in Particle 2, an antivortex is formed between two neighboring vortices of identical orientation (both are counterclockwise in this case) to create an energetically stable configuration, as observed in other ferroelectric[54,55], ferromagnetic[56], and superconducting[57] systems. Our 3D polarization measurements also provide the out-of-plane components of the local polarizations at the vortex cores, which show clear out-of-plane components along the

[100] direction at the vortex cores of both particles (Supplementary Figs. 14–16), indicating that the vortices are chiral. The normalized helicity density ($H_n$) maps also clearly show non-zero helicity density values near the vortex centers, confirming their chirality (Supplementary Fig. 17). The helicity density map further allows us to determine the handedness of each chiral vortex by examining its sign[58]. Positive helicity density can be clearly observed for all the vortices appearing in both Particles, indicating that all the vortices are right-handed, which is also evident from the fact that the direction of the curl is parallel to the axial component of the polarization (Supplementary Figs. 14–17). The antivortex and hedgehog-type structures are expected to show zero helicities in ideal configurations[58], consistent with our experimental findings (Supplementary Fig. 17). Although the existence of vortices and antivortices, along with their chirality/handedness, can be qualitatively analyzed based on the curl of in-plane polarization and helicity density maps, these methods cannot quantitatively provide polar topological information of measured 3D polarization distributions. Therefore, we further calculated topological invariants (e.g., winding number, skyrmion number, and Hopf invariant) with the circle-equivalent and sphere-equivalent order-parameter spaces[58] (Methods). First, we calculated the topological invariants defined for 2D polarization vector fields (2D vectors in a 2D plane). From the 2D polarization vector fields sliced from our 3D polarization distributions along [100] direction, local winding numbers were calculated by taking a line integral of the change in the arguments of the in-plane polarization vectors over a closed path[58]. As can be seen in Fig. 5a, b, we successfully identified the positions of vortices (winding number of +1), convergent hedgehog-type structures (winding number of +1), and antivortices (winding number of −1), along with their 3D distributions, which are consistent with the qualitative findings described above. Second, we continued to calculate the topological invariants defined for 3D vectors in a 2D plane, starting from the Pontryagin charge density maps, again for the sliced polarization distributions along the [100] direction (Methods). Figure 5c, d shows the presence of clearly finite Pontryagin charge density near the vortex and antivortex areas, indicating that each vortex and antivortex possesses a non-trivial polar topology. To achieve the complete topological classification, we further determined the skyrmion numbers for each vortex and antivortex by taking a local 2D surface integral of the Pontryagin charge density[59,60], as shown in Supplementary Fig. 18. In the case of Particle 1, the skyrmion number was found to be approximately +0.3 near the vortex core region, which can be classified as a Bloch-type fractional skyrmion. For Particle 2, as observed in the first representative slice of Supplementary Fig. 18b, the skyrmion numbers of the two vortices were +0.7 and +0.1, respectively, again characterizing them as Bloch-type fractional skyrmions. The skyrmion number of the antivortex, only found in Particle 2, was also identified as −0.4. Additionally, in the case of Particle 2, we confirmed the presence of a polar topology classified as a convergent hedgehog-type fractional skyrmion with a skyrmion number of +0.3 (Supplementary Fig. 18b). The topological analyses described above require 2D slicing of the 3D polarization distributions, and the results can vary depending on the choice of slicing direction. Moreover, due to the finite size and asymmetric geometry of the system, the integration ranges for skyrmion number calculations are limited, and occasionally, two or more neighboring topological objects coexist within the integration range. The electrostatic boundary conditions of our experimental system can also be non-ideal, exhibiting partial screening. The observation of topological charge values deviating from integer or half-integer values, which are generally not allowed in a 2D continuum model, can be attributed to these issues. Note that the emergence of such fractional topological charges under non-ideal conditions was found in previous experiment[61].

Furthermore, we observed that the skyrmion number undergoes an abrupt sign change (from −0.5 to +0.6) through the region marked

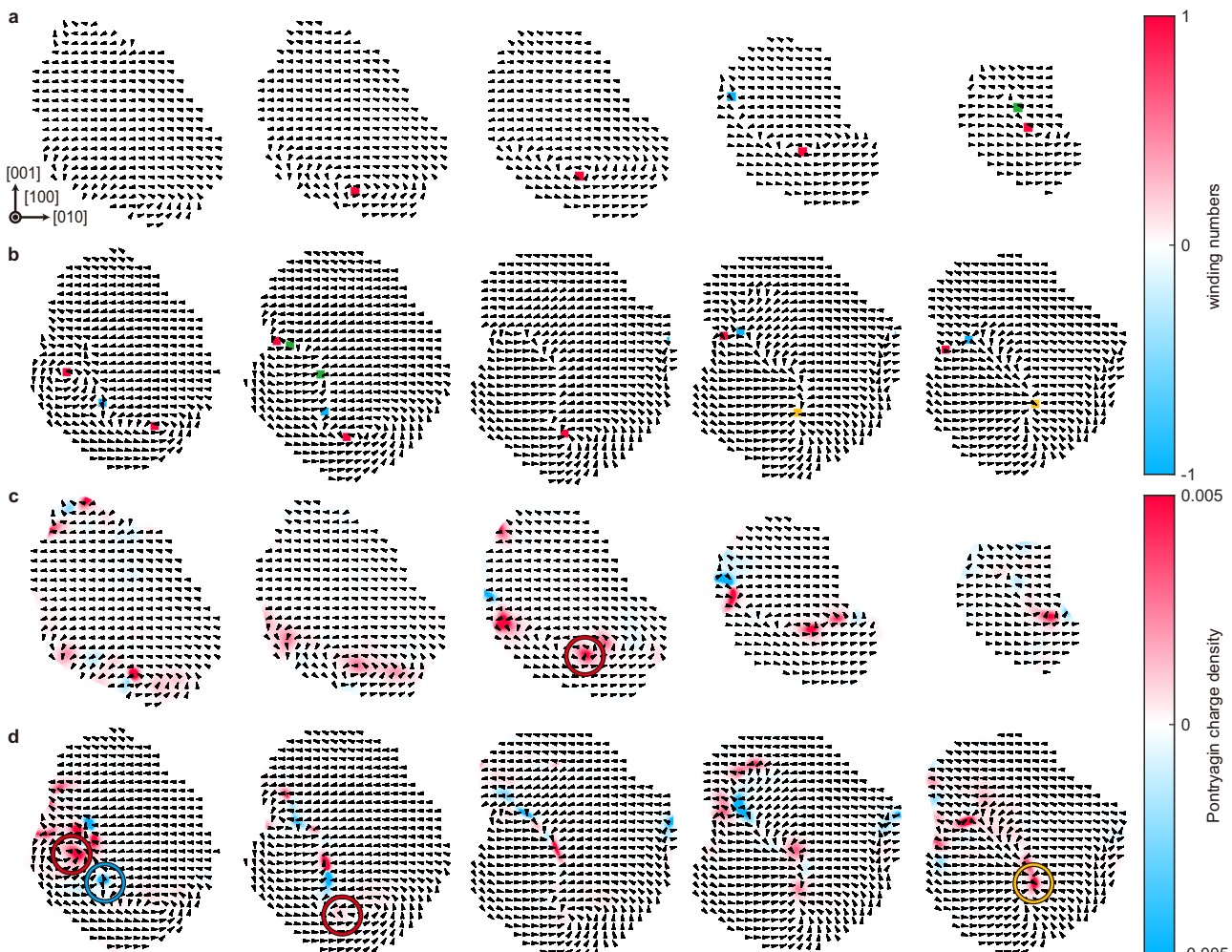

**Fig. 5 | Representative 2D slices through the nanoparticles showing topological invariants. a, b** Winding number maps of representative Ti atomic layers along the [100] direction of Particle 1 (8.8 nm) (**a**), and Particle 2 (10.1 nm) (**b**). The in-plane directions of the Ti displacements are indicated by black arrows. The red, yellow, and blue colors represent the vortex (winding number of +1), hedgehog-type structure (winding number of +1), and antivortex (winding number of −1), respectively. Note that the winding number itself cannot distinguish between stripe domains, flux-closure domains, hedgehog-type structures, and vortex (or antivortex) structures; the identification of vortex structure requires more information

regarding the local polarization distributions. Accordingly, we marked in green the areas where the winding number is +1 or −1, but which are neither vortices, hedgehog-type structures, nor antivortices. The distance between the arrows is 3.75 Å. **c, d** Pontryagin charge density maps of representative Ti atomic layers along the [100] direction of Particle 1 (8.8 nm) (**c**), and Particle 2 (10.1 nm) (**d**). The in-plane directions of the Ti atomic displacements are indicated by overlaid black arrows. Vortices, antivortices, and hedgehog-type structures are marked with red, blue, and yellow circles, respectively. The distance between the arrows is 3.75 Å.

with black squares in Supplementary Fig. 18b. This typically indicates the existence of a Bloch point, where the polarization vanishes[62]. The 3D polarization distribution near the Bloch point is shown in the Supplementary Fig. 18c, and the topological charge for the surrounding area is +1, indicating that the Bloch point structure has a diverging spiral configuration (Methods). Finally, since we have obtained the full 3D polarization distributions, the Hopf invariant[58], indicative of the integer topological charge of a hopfion (3D counterparts of skyrmions), can be calculated. Particles 1 and 2 showed the Hopf invariant of −0.002 and −0.011, respectively, indicating that neither of the particles is likely to possess a hopfion. Through these analyses we were able to observe topologically non-trivial 3D polar structures at the atomic scale and quantitatively calculate their topological invariants to classify them into different categories including fractional skyrmions and spiral Bloch point structures.

The current implementation of AET can only resolve cation atoms with relatively large electron scattering cross-sections. However, to fully understand the surface boundary condition, as well as the resulting 3D polarization distribution and underlying physical

mechanism, precise information regarding the 3D locations of low-$Z$ surface adatoms (such as carbon) and internal oxygen atoms is required. With the development of the four-dimensional STEM (4D-STEM) scheme based on high-speed pixelated electron detector technology, it is now possible to fully utilize the information contained in scattered electron beams during STEM measurement[63,64]. Several experimental and computational studies suggest that the 3D atomic coordinates of both low-$Z$ and high-$Z$ elements can be accurately measured together via electron tomography based on 4D-STEM[65–67]. We foresee that the full 3D atomic structures (including oxygens and other low-$Z$ elements on the surface) and underlying mechanisms of ferroelectric orderings in nanosized systems will be deciphered in the near future. Additionally, a 4D-STEM-based multislice electron tomography algorithm[66] allows measurements to be taken with the sample in the out-of-focus state, overcoming the limitation imposed by the depth of focus on the sample size. Since the multislice-based method can also compensate for the nonlinear effects caused by multiple scatterings, we anticipate that it will allow us to reveal the toroidal orderings in much larger nanoparticles and

the transition mechanisms from the vortex states towards the bulk polar states.

Lastly, it has been predicted that storing information as the handedness of the toroidal ordering can substantially enhance the density of stored information by up to five orders of magnitude (due to the greatly suppressed cross-talk between the information carriers)[15]. Moreover, due to the unique topological nature, polar skyrmions are expected to show enhanced stability over time, and their distinct topological protection renders them highly metastable in environments with field polarization with minimal energy dissipation[68]. If combined with the PTMO state we observed, our observation of non-trivial polar structures (fractional skyrmions and a spiral Bloch point structure) in 3D ferroelectric nanostructures without epitaxial strain can open a door towards even higher capacity storage devices, since it allows the doubling of the memory capacity by separately controlling the linear polarity and chirality, which can carry a bit of information[37]. We hope that our approaches will provide a more comprehensive understanding towards the nature of 3D polar orderings in strain-free nanostructures, and open a pathway leading to functional devices along with further development of switching and reading schemes.

## Methods

### Sample preparation

The $BaTiO_3$ nanoparticles were fabricated using the hydrothermal synthesis. For this work, $BaTiO_3$ nanoparticles were donated by Samsung Electro-Mechanics Co. Ltd. The resulting $BaTiO_3$ nanoparticles were dispersed in ethanol with 3 h of bath sonication and drop-casted onto SiN membranes of 5 nm thickness. To prevent electron beam damage and charge accumulation, an amorphous carbon film of approximately 2 nm thick was coated on the surface of the specimen. The grids were subsequently heat-treated in a vacuum at 150 °C for 10 h to minimize hydrocarbon contamination during TEM imaging.

### Initial characterization

The atomic structure and chemical composition of the nanoparticles were initially characterized using two techniques: EDS mapping with an FEI Titan cubed G2 60-300 double Cs-corrected and mono-chromated TEM at the KAIST Analysis Center for Research Advancement (KARA), and PXRD with a Rigaku SmartLab at the KARA. The EDS signals were collected by a Super-X EDS detector in ADF-STEM mode under the following conditions: 300 kV acceleration voltage, 52 pA screen current, and 512 × 512 scan points with 36.65 pm step size. For elemental mapping, Ba Lα, Ti Kα, and O Kα lines were used. The ADF image and EDS maps revealed alternating columns of Ba and Ti atoms, consistent with the <001> zone axis of cubic $BaTiO_3$ (Supplementary Fig. 1). A PXRD measurement was conducted for the nanoparticles using the Cu Kα$_1$ source (8.04 keV) in the two-theta angular range of 10° to 80° with a 0.001° angle step utilizing a D/teX Ultra 250 detector. By comparing the peaks of the PXRD pattern with the standard reference card (PDF #01-084-9618), the powder exhibited a long-range cubic phase (i.e., no (002) peak splitting and $c/a$ of unity), but the precision of our ($c/a$) value was estimated to be 0.006, considering the PXRD peak widths (Supplementary Fig. 1g). We cannot rule out the possibility that our particles have tetragonal distortion within this $c/a$ ratio range. The lattice constant of a five-atom unit cell (pseudocubic cell) was determined to be 4.026 ± 0.001 Å based on the angular positions of (211), (220), (300), (310), and (311) diffraction peaks.

### STEM data acquisition

Two tomographic tilt-series of the $BaTiO_3$ nanoparticles were acquired using an FEI Titan Themis3 double Cs corrected and monochromated TEM at the Korea Basic Science Institute in Seoul (for Particle 1), and an FEI Titan cubed G2 60−300 double Cs corrected and monochromated TEM at the KARA (for Particle 2), respectively. The images for each tilt-series were obtained using the ADF-STEM mode at 32 different tilt

angles, ranging from −73.0° to +71.6° for Particle 1 and −73.8° to +71.6° for Particle 2, respectively (Supplementary Fig. 2). All the images were acquired under the following microscope parameters: acceleration voltage of 300 kV, detector inner and outer semi-angles of 38 mrad and 200 mrad, respectively, convergence beam semi-angle of 18.0 mrad, and beam current of 7 pA. At each tilt angle, three consecutive images of 1024 × 1024 pixels were collected using a pixel dwell time of 3 μs with a pixel size of 0.354 Å for Particle 1 and 0.364 Å for Particle 2. Note that the pixel sizes were calibrated to match the lattice constant determined from the PXRD with the 3D structures obtained from electron tomography analysis. The total electron dose for the entire tilt series was $1.01 \times 10^5$ $e$ Å$^{-2}$ for Particle 1 and $9.52 \times 10^4$ $e$ Å$^{-2}$ for Particle 2, respectively. Existing studies show that the electron beam exposure of up to $8.78 \times 10^5$ $e$ Å$^{-2}$ does not alter the cation displacement of a $BaTiO_3$ nanocluster of approximately 4 × 4 nm$^2$ size even at an elevated temperature of 473 K[38]. Therefore, no significant structural damage is expected under the electron dose used in this study. Furthermore, the domain formation or switching induced by electric fields resulting from charge accumulation due to the electron beam is also not a significant concern in this study, as it requires at least ten times larger electron dose[69]. To ensure minimal beam-induced structural changes during the tilt series acquisition, the zero-degree projections were measured three times throughout the experiment (at the beginning, in the middle, and at the end of the tilt series acquisition). Supplementary Fig. 3a−f illustrates that the internal structures observed in the experiments exhibit consistency. While mild beam-induced atomic diffusion can be observed at the surfaces, particularly for the smaller nanoparticle (Particle 1), this phenomenon was expected and considered during our analysis. The signal from the diffused surface atoms is likely to be averaged out during the reconstruction process[31], and thus its impact on the reconstructed internal structure becomes insignificant. We artificially introduced surface amorphization to a known structure and simulated a tomography experiment on the changing surface structure to check the effect of surface diffusion. This indeed verified that its effect on the reconstructed internal structure is negligible compared to the precision of AET (More details regarding the evaluation of the effect of surface diffusion are described in later sections of Methods below).

### Image post-processing and 3D reconstruction

We performed image post-processing for the tilt series using a series of procedures, including drift correction, scan distortion correction, noise reduction[70,71], background subtraction, and tilt-series alignment, following the methods[28,29,31,72–74].

(I) Drift correction. We applied drift correction to the images collected at each tilt angle. The stage drift during acquisition was estimated from the three consecutively measured ADF-STEM images and subsequently compensated for using an affine transform that inverts the estimated linear drift effect.

(II) Scan distortion correction. We corrected the ADF-STEM scan distortion resulting from the miscalibrations of fast and slow scan directions by measuring the distortion matrix using a single crystal silicon (110) standard sample. The scan distortions were corrected by applying the proper shear and scaling operations corresponding to the distortion matrix in the Fourier domain[75].

(III) Image denoising. To address the Gaussian-Poisson mixed noise in ADF-STEM images, we first estimated the Gaussian-Poisson noise parameters from a statistical analysis based on the three consecutively acquired images. The images were then denoised using the block-matching and 3D filtering (BM3D) algorithm[70]. To convert Gaussian-Poisson noise into pure Gaussian noise, as required by the BM3D algorithm, we applied the Anscombe transformation[71] and its inverse transformation to the images before and after the BM3D de-noising process with the estimated noise parameters, respectively.

(IV) Background subtraction. After denoising, we defined a mask slightly larger than the boundary of the nanoparticles for each projection image. We determined the background inside the mask by solving the Dirichlet boundary value problem of the discrete Laplace's equation, and subtracted it from the image.

(V) Tilt-series alignment. Each tilt series was then aligned based on the center-of-mass[76] and common-line alignment[28] methods with sub-pixel accuracy. The images were shifted to a consistent center-of-mass position along the direction perpendicular to the tilt axis. Then, the common-line was computed from each image by projecting it onto the tilt axis. The tilt series was subsequently aligned along the direction parallel to the tilt axis by shifting the projections to ensure complete overlap of the common lines.

(VI) 3D reconstruction. After the image post-processing, we reconstructed 3D tomograms from the post-processed tilt series using the GENFIRE algorithm[32] with the following parameters: discrete Fourier transform (DFT) interpolation method, number of iterations of 500, oversampling ratio of 4, and interpolation radius of 0.1 pixels.

## Atom tracing and species classification

The 3D atomic coordinates of internal cation atoms in each nanoparticle were determined using the following atom tracing procedures[28,72,77].

(i) The 3D local maxima positions in the tomograms were identified and sorted based on the peak intensity in descending order. Starting from the highest intensity, a volume of $1.8 \times 1.8 \times 1.8 \, \text{Å}^3$ ($5 \times 5 \times 5$ voxels) centered on each local maximum was cropped, and a 3D Gaussian function was fitted for the cropped volume to find the peak position with sub-pixel accuracy. The fitted peak position was added to the traced atom list only if it did not violate the minimum distance constraint of 2 Å compared to any of the fitted positions already in the list. By repeating this process for all the 3D local maxima, a list of potential atom positions was obtained.

(ii) The potential atom positions were classified into three types of atoms (non-atom, Ti, and Ba) using a *k*-means clustering algorithm[28,72,77]. With this classification procedure, we obtained an initially classified atomic model.

(iii) Due to the imperfectness of the experimental images (non-linear effect, noise, misalignment, etc.), the initial atomic structure contained some atoms that were inconsistent with the $ABO_3$ perovskite structure. As the cation atoms (Ti and Ba) of $BaTiO_3$ nanoparticles should exhibit a bcc configuration, bcc lattice fittings were applied to the initially classified atomic models to filter out the atoms not compatible with the perovskite structure. More details regarding the bcc lattice fitting procedures are described in later sections of Methods below.

(iv) The Ba and Ti layers were defined for each atomic plane perpendicular to the <001> direction, based on the initial tracing and classification results. First, histograms were drawn based on the atom positions along the reference directions ([010] direction for Particle 1 and [001] direction for Particle 2), and the atomic layers along the reference directions were defined based on the peak positions in the histograms. The atomic positions were then assigned to corresponding atomic layers of the smallest distance. The layers dominated by Ba atoms were identified as the Ba layers, while the layers with dominant Ti atoms were identified as the Ti layers.

(v) Due to the intensity elongation effect of the tomogram along the missing-wedge direction and slight imperfections caused by the data imperfectness described above, several connected intensity blobs appear in the tomogram. Since the 3D local maxima of the connected intensity blobs are not well-defined, an additional atom tracing procedure was necessary to fully determine the atomic structures. The 3D tomogram was sliced [slice thickness of 1.1 Å (3 pixels)] for every atomic layer in either the [010] direction (for Particle 1) or the [001] direction (for Particle 2), and each slice was summed along the slice

directions to obtain 2D images. For all the 2D slices, the positions and intensities of 2D local maxima were extracted. Starting from the highest intensity, the 3D Gaussian fitting procedure was repeated using the 2D local maxima positions as initial positions. The size of the cropped volume for the 3D Gaussian fitting was adjusted by changing the side length, ranging from three to seven pixels. We then added the new best-fit position, obtained from the volume size showing the smallest mean squared residuals, to the traced atom list, provided that the minimum distance constraint of 2 Å between neighboring atoms was met. We further applied bcc lattice fittings again to eliminate some of the newly added fit positions that did not conform to the perovskite structure.

(vi) To finalize the 3D atomic structures, we manually removed physically unreasonable atom candidates and added candidates that were missed during the automated process. During this process, we used a minimum distance constraint of 2 Å between the nearest atoms. In total, we manually added and removed 406 and 39 atoms (for Particle 1) and 265 and 28 atoms (for Particle 2), respectively, resulting in the final 3D cationic atomic models of 11,126 atoms (Particle 1) and 17,260 atoms (Particle 2). Note that the manual adjustment of this level has been commonly employed in the field of AET[28,77].

(vii) After manually adjusting the atomic positions, we classified the final atomic structures into Ba and Ti atoms based on the layer labellings determined in step (iv), producing 3D models of 11,126 atoms with 5497 Ti and 5629 Ba atoms for Particle 1 and 17,260 atoms with 8268 Ti and 8992 Ba atoms for Particle 2, respectively.

(viii) To assess the consistency of the determined atomic structures with the measured data, we calculated the R-factors[28,29,73] by comparing the experimentally acquired tilt series with the simulated tilt series forward-projected from the final 3D atomic models. The averaged R-factors were 0.22 for Particle 1 and 0.17 for Particle 2. Given that R-factor minimization[28,72] was not carried out during the structural determination and only cation atoms were used for the forward-projection calculation (excluding oxygens), the R-factors we obtained can be seen as indicative of the reliability of our atomic structures. In fact, R-factors in the range of 0.20–0.25 are generally considered acceptable in the crystallography community[78–81].

## Assignment of experimental 3D atomic positions to ideal bcc lattices

Since the cation atoms of $BaTiO_3$ nanoparticle display a bcc configuration, the 3D atomic positions obtained from the tomograms were assigned to ideal bcc lattice sites using the following procedures.

(a) First, histograms were drawn based on the atom positions along the reference directions ([010] direction for Particle 1 and [001] direction for Particle 2), and the atomic layers along the reference directions were defined based on the peak positions in the histograms.

(b) Each atom was classified into the atomic layer with the smallest distance to it.

(c) A Ba atom closest to the mean position of the 3D atomic coordinates was chosen as the origin of an ideal bcc lattice.

(d) Next, the nearest and next-nearest bcc sites of this atom were calculated based on the initial pseudocubic lattice vectors with a lattice constant of 4.036 Å (the bulk lattice constant of cubic $BaTiO_3$).

(e) For each calculated site, if an atom is found within 25% of the nearest neighbor distance of the bcc lattice (initially 0.87 Å), the component of the lattice vector (i.e., the vector from the origin to the calculated site) along the reference direction was compared with the atomic layer assigned to the found atom. If the position of the atomic layer (relative to the origin) is consistent with the lattice vector component, the candidate atom was assigned to that bcc lattice site.

(f) The nearest and the next-nearest neighbor search was repeated for all the newly assigned bcc lattice sites. This process was repeated until no further atoms could be assigned to the lattice.

(g) New bcc lattice vectors were fitted to the atoms assigned to the lattice using fitting parameters of translation, 3D rotation, and lattice constant parameters, to minimize the error between the measured atomic positions and the corresponding lattice sites of the fitted bcc lattice.

(h) The iterative processes from step (c) to (g) were repeated until there were no further changes in the fitted lattice vectors. After this procedure, 99.63% (Particle 1) and 99.97% (Particle 2) of the target atoms were successfully assigned to the bcc lattices. The root-mean-square deviations (RMSDs) between the assigned atom positions and the fitted bcc lattices were 92.5 pm (Particle 1) and 75.1 pm (Particle 2), respectively.

## Local tetragonal lattice fitting

For the analysis of local tetragonality ($c/a$ ratio), we performed a local tetragonal lattice fitting using the following procedures. First, for each Ti atom, we identified eight nearest-neighbor Ba atoms based on the globally fitted bcc lattice configuration (see 'Assignment of experimental 3D atomic positions to ideal bcc lattices' section of Methods). Second, we performed three tetragonal lattice vector fittings to the atoms assigned to the lattice, for [100], [010], and [001] as potential $c$-axis directions, respectively. This fitting involved parameters for translation, 3D rotation, and scaling, and was done to minimize the error between the measured atomic positions (which includes only the 8 nearest neighbor atoms for each Ti atom) and the corresponding lattice sites of the fitted tetragonal crystal lattice. Among these, the $c$-vector direction that minimized the errors between the measured atomic positions and the corresponding lattice sites of the fitted tetragonal crystal lattice was determined as the $c$-axis for that particular local structure. This procedure was applied to all Ti atoms which have eight nearest neighbor atoms. As a result, the average $c/a$ ratio for each nanoparticle was determined to be $1.002 \pm 0.002$ for Particle 1 and $1.002 \pm 0.001$ for Particle 2. Note that the standard error propagation, based on the precision of the atomic coordinates, was used to determine the error bars in this process.

## Precision analysis using multislice simulation

To evaluate the reliability of our final 3D atomic structures of the two BaTiO$_3$ nanoparticles, we conducted a precision analysis. From the determined 3D atomic structures for Particle 1 and Particle 2, 32 projection images were calculated using multislice simulation[82,83] at the experimental tilt angles under the following parameters: incident electron energy of 300 keV, convergence semi-angle of 18 mrad, detector inner and outer semi-angles of 38 mrad and 200 mrad, respectively, slice thickness of 2 Å, −775 nm C$_3$ and +378 μm C$_5$ aberration for Particle 1, and 130 nm C$_3$ and 0 μm C$_5$ aberration for Particle 2. Eight frozen phonon configurations were used to consider the thermal vibration effect at room temperature, which corresponds to the random spatial displacement with a standard deviation of 7.4 pm for Ti and 5.6 pm for Ba[36]. To consider the impact of the electron probe size and other incoherent effects, we applied a Gaussian kernel with a standard deviation of 0.46 Å to each multislice simulated image.

We then employed the GENFIRE algorithm[32] to reconstruct the 3D tomograms from the multislice-simulated tilt series by using the following parameters: DFT interpolation method, number of iterations of 500, oversampling ratio of 4, and interpolation radius of 0.3 pixels.

The atomic structures were determined from the 3D tomograms using the same methods described above (see above for more details regarding atom tracing and species classification). To compare the experimentally determined atomic structures with those from multislice simulations, we calculated the distances between atoms in the experimental structure and the multislice-simulated structure. Any pairs of atoms whose distance was less than a specified threshold distance (half of the nearest-neighbor distance of the ideal fitted bcc lattice: 1.7 Å) were considered as common atom pairs. The result showed that 90.2% of the atoms in Particle 1 and 94.0% of the atoms in Particle 2 were properly retrieved in the simulated experiments. The RMSDs of all common atom pairs (i.e., the precision of the atomic coordinates; see refs. [28,31,72]) were found to be 38.3 pm (Particle 1) and 34.4 pm (Particle 2), respectively.

## Nanoparticle size estimation

To estimate the size of the BaTiO$_3$ nanoparticle, the volume of the nanoparticle was calculated as the product of the number of unit cells (estimated as the number of Ti atoms) and the volume of a pseudo-cubic unit cell (65.5 Å$^3$). If we assume a spherical shape, the diameters of the nanoparticles can be estimated to be 8.8 nm for Particle 1 and 10.1 nm for Particle 2, respectively.

## Ti atomic displacement field and polarization field calculation

The Ti atomic displacements from the centrosymmetric position of each unit cell were calculated as the deviation of each 3D Ti position from the geometric center of the eight neighboring Ba atomic positions. The neighboring Ba atoms for each Ti atom were determined as the Ba atoms assigned to the nearest neighbor sites around the Ti atom based on the fitted bcc lattices. Furthermore, the calculation of Ti atomic displacement was performed only for Ti atoms that had all eight nearest neighbor Ba atoms.

In order to calculate the local polarization distribution of BaTiO$_3$[40,44], the Ti atomic displacement of each unit cell and the displacement of the oxygen octahedron must be identified, as shown in Eq. (1):

$$P_s = \kappa\left(\delta_{Ti} - \delta_O\right). \tag{1}$$

Here, $\kappa$ is an empirical constant with a value of $1.89 \pm 0.15$ (C m$^{-2}$) Å$^{-1}$, obtained using the measured spontaneous polarization and the displacement of the Ti atom from the center of the oxygen octahedron in bulk BaTiO$_3$[40]. $\delta_{Ti}$ and $\delta_O$ are the Ti atomic displacement and the displacement of the oxygen octahedron with respect to the centrosymmetric position of each unit cell, respectively. Since current AET methods can only identify cation atoms, an alternative expression for spontaneous polarization[84] using structural information was additionally employed. This expression is given by

$$P_s = \frac{e}{V}\left(2Z^*_{O\perp}\delta_O + Z^*_{O\parallel}\delta_O + Z^*_{Ti}\delta_{Ti}\right), \tag{2}$$

where $V$ is the volume of a unit cell, $Z^*_{O\perp}$, $Z^*_{O\parallel}$, and $Z^*_{Ti}$ are the Born effective charges of O perpendicular to the Ti-O direction, O parallel to the Ti-O direction, and Ti atom, respectively. The Born effective charges of each case ($Z^*_{O\perp} = -2.13$, $Z^*_{O\parallel} = -5.75$, and $Z^*_{Ti} = 7.29$) we used in this study were obtained from ref.[85], which were calculated through ab initio theory with a cubic phase BaTiO$_3$ having a lattice constant of 4.00 Å.

By integrating Eqs. (1) and (2) to express $\delta_O$ as $\delta_{Ti}$, and using the value of $\kappa$, $Z^*$, and $V$, spontaneous polarization can be calculated by simply multiplying the experimentally observed $\delta_{Ti}$ with the scale factor ($\alpha$):

$$P_s = (2.12 \pm 0.69)\delta_{Ti} = \alpha\delta_{Ti}. \tag{3}$$

To obtain the Ti atomic displacement field, we first interpolated the Ti atomic displacements ($\delta_{Ti}$) onto a 3D Cartesian grid with 0.25 Å pixel size by applying a 3D Gaussian kernel[31,86] with a standard deviation of 4.03 Å, equivalent to the nearest neighbor distance between Ti atoms. Subsequently, the polarization field was obtained by multiplying the scaling factor ($\alpha$) between Ti atomic displacement ($\delta_{Ti}$) and the spontaneous polarization ($P_s$). The standard error propagation, based on the precision of the atomic coordinates and the $\kappa$ value, was used to

determine the error bars in this process, as well as in other calculations conducted in this study.

Note that we applied kernel averaging to atomic displacement vectors and polarization to increase the precision of the individual displacement vectors. Regardless of kernel averaging, the polarization distribution exhibits substantial vortex structures (Supplementary Figs. 6 and 8). To further confirm whether the polarization distribution maintains the same vortex state prior to Gaussian kernel averaging, we calculated the winding number both before and after kernel averaging, as shown in Supplementary Fig. 19. These results indicate the consistent presence of substantial, non-zero winding numbers in vortex regions, even before the application of kernel averaging.

### Analysis of swirling characteristics of the polarization vector fields

A vortex is defined by the rotation of polarization around a core, and its rotational behavior can be described by a toroidal moment ($\mathbf{G}$), curl of in-plane polarization, and chirality. The toroidal moment[15,87], which is determined by the cross product of local polarization with their atomic positions, is given by

$$\mathbf{G} = \frac{1}{2N} \sum_i \mathbf{r}_i \times (\mathbf{p}_i - \mathbf{P}), \qquad (4)$$

where $N$ is the number of unit cells, $\mathbf{r}_i$ is the atomic position of the $i^{\text{th}}$ Ti atom, $\mathbf{p}_i$ is the local polarization of the $i^{\text{th}}$ Ti atom (before kernel averaging), and $\mathbf{P}$ is the averaged polarization (before kernel averaging).

Curl of in-plane polarization (also called as axial current[88]) is a concept in fluid mechanics that describes the rotational motion of fluid around a common centerline[89]. In the context of polar structures, it is also used to characterize the clockwise or counterclockwise toroidal ordering[23,90,91]. The curl of the in-plane polarization map was obtained by slicing the 3D polarization vector field along the [100] direction and then calculating the $x$-component of the curl for the normalized in-plane polarization fields. Note that, in this study, the in-plane polarization fields were normalized before the curl of in-plane polarization calculation since the unnormalized polarization approaches zero in the vicinity of the vortex center, resulting in the emergence of a singularity.

Chirality can be determined through the normalized helicity density ($H_n$), a mathematical concept adapted from fluid dynamics, which is defined as[92]

$$H_n = \frac{\mathbf{P} \cdot (\nabla \times \mathbf{P})}{|\mathbf{P}||\nabla \times \mathbf{P}|}, \qquad (5)$$

where $\mathbf{P}$ is local polarization.

### Topological structures analysis

The characteristics of topological structures can be described by topological invariants (e.g., winding number, skyrmion number, and Hopf invariant). Winding number is defined as a topological invariant in the circle-equivalent order-parameter space, and is used to distinguish between vortices (winding number of +1) and antivortices (winding number of −1) in a 2D system. The winding map was calculated by taking a line integral of the change in the arguments of the in-plane polarization vectors over a closed path (square loop with a side length of 3.75 Å) near different positions in the 2D polarization vector fields (2D vectors in a 2D plane) sliced from our 3D polarization distributions along [100] direction[93].

Skyrmion number ($N_{sk}$) is defined as a topological invariant in the sphere-equivalent order-parameter space and is used to characterize the swirling structure of a skyrmion. To obtain skyrmion number, we first calculated the Pontryagin charge density ($q_x$) maps for the 2D

polarization vector fields (3D vectors in a 2D plane) sliced from our 3D polarization distributions along the [100] direction (Fig. 5c, d). This expression is given by[59,60]

$$q_x = \frac{1}{8\pi} \epsilon_{ijk} \widehat{\mathbf{P}} \cdot \left( \partial_j \widehat{\mathbf{P}} \times \partial_k \widehat{\mathbf{P}} \right) \bigg|_{j=y,k=z} = \frac{1}{4\pi} \widehat{\mathbf{P}} \cdot \left( \partial_y \widehat{\mathbf{P}} \times \partial_z \widehat{\mathbf{P}} \right), \qquad (6)$$

where $\epsilon_{ijk}$ is the Levi–Civita symbol, and $\widehat{\mathbf{P}}$ is the unit vector representing the direction of local polarizations. For the topological characterization of 3D polarization vector fields, we further calculated the skyrmion numbers for each vortex and antivortex, which are given by

$$N_{sk} = \int q_x dA_x = \int \frac{1}{4\pi} \widehat{\mathbf{P}} \cdot \left( \partial_y \widehat{\mathbf{P}} \times \partial_z \widehat{\mathbf{P}} \right) dy dz. \qquad (7)$$

The skyrmion map was determined by performing a local 2D surface integral of the Pontryagin charge density over a circular area with a radius of 10.00 Å for Particle 1 and 13.75 Å for Particle 2, centered around each position in the (100) plane (Supplementary Fig. 18). Note that skyrmion numbers at each vortex and antivortex can vary depending on the chosen integration areas. Therefore, we calculated multiple skyrmion numbers at the locations of vortices, antivortices, and hedgehog-type structures for several different choices of the integration area and checked their trends. These trends clearly indicate that the skyrmion numbers saturate beyond certain integration areas, which were chosen as the area for our analysis given above. To calculate the topological charge ($Q$) at the Bloch point, the left-hand side of Eq. (7) can be generalized into 3D based on the divergence theorem[94].

$$Q = \int q_x dA_x = \int \frac{3}{4\pi} \partial_x \widehat{\mathbf{P}} \cdot \left( \partial_y \widehat{\mathbf{P}} \times \partial_z \widehat{\mathbf{P}} \right) dx dy dz. \qquad (8)$$

Hopf invariant ($N_H$), representing the integer topological charge for a hopfion (3D counterparts of skyrmions), was calculated using the following equation[58,95].

$$N_H = - \int \mathbf{F} \cdot \mathbf{A} dV. \qquad (9)$$

Here, $\mathbf{F}_i = \frac{1}{8\pi} \epsilon_{ijk} \widehat{\mathbf{P}} \cdot (\partial_j \widehat{\mathbf{P}} \times \partial_k \widehat{\mathbf{P}})$ and $\mathbf{A}$ is defined as a gauge field satisfying $\mathbf{F} = \nabla \times \mathbf{A}$. The vector potential $\mathbf{A}$ was determined in momentum space using the Coulomb gauge condition ($\nabla \cdot \mathbf{A} = 0$), and based on this, the Hopf invariant was calculated in momentum space[95].

### Evaluation of the effect of possible surface diffusion during tilt series measurement via simulation

By comparing the experimental projections at zero-degree before and after tomography measurement (Supplementary Fig. 3a–c), we could observe mild structural changes near the surface, especially for Particle 1. To evaluate the effect of surface diffusion for the determination of internal polarization structures, we simulated 3D reconstruction considering the surface diffusion effects and calculated the precision (RMSD) of the retrieved internal structure.

From the determined 3D cation atomic structure of the Particle 1 (total 11,126 atoms), 5,452 atoms were selected as surface atoms by repeatedly applying the Alpha-shape algorithm[96] with shrink factor 1.

To simulate the effect of surface diffusion due to the electron beam damage, the position of each surface atom was randomly perturbed, resulting in the RMSD of 300 pm when we compare the structure before and after the random perturbation (Supplementary Fig. 3g–i). A tilt series of 32 projections was generated by linearly projecting a 3D potential volume from the determined cation atomic structure of Particle 1 (we named this Tilt-Series S1) along the

experimental tilt angles. Another tilt series was similarly generated, but for this tilt series, half of the projections were randomly chosen and replaced with the projections obtained from the structure perturbed by the random surface displacement (we named this Tilt-Series S2).

Two 3D reconstructions were obtained from the tilt series (Tilt-Series S1 and S2) using the GENFIRE algorithm[32] with the following parameters: DFT interpolation method, number of iterations of 500, oversampling ratio of 4, and interpolation radius of 0.3 pixels. Using the same atom tracing method, two traced 3D atomic models (model S1 and S2) were obtained from the two 3D reconstruction volumes, respectively. Although the structures of the surface atoms were very different between the two models (due to the perturbation of 300 pm RMSD we applied), the structures of the internal atoms (i.e., non-surface atoms) were well-preserved and the RMSDs of the atom pairs between the internal atoms in the experimental 3D atomic structure and the traced atomic models (model S1 and S2) were calculated to be 27.3 pm and 39.0 pm, respectively, comparable to the precision our technique can provide.

We subsequently performed the bcc lattice fitting and Ti atomic displacement field analysis (using the same methods described in the above sections) for the internal atoms (i.e., non-surface atoms) of the traced atomic models (model S1 and S2). As can be seen in the Supplementary Fig. 3j–l, consistent internal vortex structures could be obtained even from the tomographic reconstruction which suffered from the surface diffusion effect.

### Effective Hamiltonian calculations

The effective Hamiltonian method described in ref. [87] was employed to calculate the local polarization distributions of ellipsoidal $BaTiO_3$ nanoparticles whose sizes are similar to the experimentally measured ones. In this simulation, the coefficient in front of the square of the local mode within the self-mode energy was slightly changed to place the computational critical temperature of the tetragonal-to-orthorhombic phase transition of bulk $BaTiO_3$ to the experimental value of 278 K[97]. This ensures that the room-temperature equilibrium phase of the $BaTiO_3$ bulk is tetragonal. This Hamiltonian has three degrees of freedom, namely the local mode at each 5-atom site (which is directly proportional to the local electric dipoles at these sites), and the homogeneous and inhomogeneous strains. Two different sizes were considered in line with our experiments. The first particle (Particle $\tilde{1}$) has the dimensions of (23, 22, 21) in terms of primitive 5-atom unit cells along the $x$, $y$, and $z$-directions, respectively (chosen to be along the [100], [010], and [001] pseudocubic directions). The second particle (Particle $\tilde{2}$) is slightly larger, with dimensions of (23, 24, 26). The pseudocubic lattice parameter is 4.01 Å for both Particle $\tilde{1}$ and Particle $\tilde{2}$ at 300 K. The corresponding diameter of each particle used in the simulation was estimated by taking the geometric mean of the dimensions along each axis, resulting in values of 8.8 nm (Particle $\tilde{1}$) and 9.7 nm (Particle $\tilde{2}$), respectively.

During the simulation, the samples were gradually cooled from 600 K within a Monte-Carlo procedure under mechanically and electrically free boundary conditions, while the 3D local polarization distributions were obtained for each temperature. The toroidal moment[87] given by

$$\mathbf{G} = \frac{Z^* e a}{2V} \sum_i \mathbf{r}_i \times (\mathbf{u}_i - \langle \mathbf{u} \rangle) \qquad (10)$$

were also calculated at each temperature, where the dipole vortex has started to form around 300 K (Supplementary Fig. 12a, b). Here, $Z^*$ is the dynamical charge, $e$ is the electron charge, $a$ is the lattice constant for the 5-atom pseudocubic unit cell, $V$ is the volume of the nanodot, $\mathbf{r}_i$ is the position vector for the $i^{th}$ unit cell, $\mathbf{u}_i$ is the local mode of the $i^{th}$ unit cell, and $\langle \mathbf{u} \rangle$ is the averaged local mode over the entire particle.

### Data availability

Data regarding Supplementary Figs. for dot/line/histogram plots are provided with this paper. All of our experimental data, tomographic reconstructions, determined atomic structures, and 3D Ti displacement vector field results are posted on a public website (https://mdail.kaist.ac.kr/PolarOrder3D), and they can also be accessed through an open repository (https://doi.org/10.5281/zenodo.10863645) upon publication.

### Code availability

Source codes are posted on a public website (https://mdail.kaist.ac.kr/PolarOrder3D), and they can also be accessed through an open repository (https://doi.org/10.5281/zenodo.10863645) upon publication.

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

## Acknowledgements

We thank Yong Jung Kim, Myoungjean Bae, Seok Jo Hong, Jaehyu Shim, Kyung-Jin Lee, Se Kwon Kim, Junhan Kim and Chan-Ho Yang for helpful discussions. This research was supported by the National Research Foundation of Korea (NRF) Grants funded by the Korean Government (MSIT) (No. 2020R1C1C1006239 and RS-2023-00208179). Y.Y. also acknowledges the support from the KAIST singularity professor program. J.S. and S.-Y.C. acknowledge the Basic Science Research Program (2020R1A4A1018935) and POSTECH-Samsung ElectroMechanics Cooperative Research Center. S.P. and L.B. thank ONR Grant No. N00014-21-1-2086, the Vannevar Bush Faculty Fellowship (VBFF) Grant No. N00014-20-1-2834 from the Department of Defense and ARO Grant No. W911NF-21-2-0162 (ETHOS). Part of the STEM experiments were conducted using a double Cs corrected Titan cubed G2 60-300 (FEI) equipment at KAIST Analysis Center for Research Advancement (KARA). Excellent support by Hyung Bin Bae, Jin-Seok Choi, Su Min Lee, and the staff of KARA is gratefully acknowledged. The tomography data analyses were partially supported by the KAIST Quantum Research Core Facility Center (KBSI-NFEC grant funded by Korea government MSIT, PG2022004-09). We declare that the authors utilized the ChatGPT (https://chat.openai.com/chat) for language editing purpose only, and the original manuscript texts were all written by human authors, not by artificial intelligence.

## Author contributions

Y.Y. conceived the idea and directed the study. C.J., J.S., K.-J.G, S.-Y.C. and Y.Y. prepared the TEM specimens. C.J., H.J., H.B. and Y.Y. designed and performed the tomography experiments. C.J. and J.O. conducted the XRD experiment. C.J., J.L. and Y.Y. conducted the experimental data analyses including topological invariant calculations. S.P. and L.B. conducted the effective Hamiltonian simulations. C.J., S.P., L.B. and Y.Y. wrote the manuscript. All authors commented on the manuscript.

## Competing interests

C.J., J.L., H.J., J.O. and Y.Y. have patent application (Korea, 10-2023-0133750), which disclose three-dimensional polarization mappings for ferroelectrics. The remaining authors declare no competing interests.
