## [Peer Review File · Nature Communications]

Revealing the Three-Dimensional Arrangement of Polar Topology in NanoparticlesREVIEWER COMMENTS

Reviewer #1 (Remarks to the Author):

Comments for the manuscript " Revealing the Three-Dimensional Arrangement of Polar Topology in Nanoparticles " by Chaehwa Jeong and colleagues for Nat. Comm.

The significance of topological defects in modern physics is so profound that we are yet to fully grasp all the potential future applications of these enigmatic self-organized local structures. The past two decades have been marked by intense exploration into various types of topological defects in magnetic systems, and more recently, the focus has expanded to include ferroelectric topology, particularly in the study of ferroelectric skyrmions.

The referred paper is devoted the 3D-polarization arrangement using atomic electric tomography (AET) in BTO nanoparticle. Such technique has been also use before to see 3D view of object in many aspects. Here the interesting part is mapping the polarization which is so far has been done by STEM based technique.

The experimental data are quite ok but lack of physics behind that. They just report the observation. It requires quite lot improvement. Here I have some concern about the manuscript.

1. If they see the vortex, where will the axial component of the polarization.
2. Can they demonstrated also real 3D image of a single vortex and anti-vortex using whole nanoparticle. At the moment it is only 2d slice.
3. In partice 1 there are only vortex-like structures while in partice 2 observed vortex and anti-vortex, why?Formation of vortex and antivortex is not clear although theory also tell they verify the experiment.
4. Author claimed, bigger particle will generate more vortex. Can they take bigger particle and do the same experiment. I am interested to see the vortice pattern over the particle. In fact, this is important for readers as well.
5. If I understand correctly, from this method one can not determine the vorticity, chirality and topological number so far. But an object is a topological or not determine by its topological number. They just give some references which I believe is not enough.
6. They claimed counterclockwise vortex exists in consecutive layers of the bottom half of the nanoparticle. But haven't discussed why only in the bottom half, what about another part of the particle that does not get a vortex-like structure?
7. Why not other particles of the same size range selected, do results reproduce in other nanoparticles or only in these 2 nanoparticles which are reported?
8. Does the vortex-like structure remain stable over time or changes
9. Also, author could update their conclusion, means, if they do such evalutory measurements on ferroelectric skyrmion domain, what will be the exciting.

Reviewer #2 (Remarks to the Author):

Identifying three-dimensional atomic morphology of topological domains and the distribution of the ferroelectric polarization in real space is of scientific and technological importance. In this manuscript, Chaehwa Jeong et. al. applied atomic electron tomography and unveiled the three-dimensional arrangement of ferroelectric polarizations in BaTiO₃ nanoparticles. They show that the polarization distribution within BaTiO₃ nanoparticles exhibits a vortex state, and the distribution of vortex is correlated with the size of the nanoparticles showing a transition from the single state to the multiple states. Although the authors showed preliminary structures of the three-dimensional vortices, there are still some limitations at the present stage. I suggest the authors address the following concerns before the work is considered for publication in Nature Communications.

1. The authors showed an average atomic displacement of Ti within the nanoparticles to be 18.8 pm. This value is much larger than the bulk value (~ 5 pm), and such a significant displacement would result in a substantial ferroelectric polarization ($\sim 100 \mu\text{C cm}^{-2}$), which is conflict with the estimated value in the manuscript.

2. The ferroelectric polarization is strongly correlated with the lattice in perovskite oxide ferroelectrics. The 18.8 pm displacement corresponds to a considerable c/a value in the BTO unit cell. However, the authors claimed that the unit cell of BTO is close to cubic.
3. The precision of the atomic coordinates is estimated to be 38.3 and 34.4 pm, which are much larger than the ideal displacement of Ti columns in BTO, indicating that such methodology may be not suitable to characterize the polarization behavior in this material. Ferroelectric with larger spontaneous polarization might be much more suitable for investigating the 3D polarization distribution.
4. The authors analyzed the ferroelectric vortex structures in BTO nanoparticles layer by layer, and the results still remain in two-dimensional planes, without any additional 3D information of the lattice ferroelectric polarization.

RESPONSE TO REVIEWERS' COMMENTS

First, we thank the reviewers for taking the time and effort to provide constructive comments, which have been very helpful in making improvements to the manuscript.

Please find below our response to the reviewers' comments on our manuscript "*Revealing the Three-Dimensional Arrangement of Polar Topology in Nanoparticles*". We have carefully addressed all the reviewers' criticisms and suggestions and trust that with the presented changes, the manuscript is now acceptable for publication in *Nature Communications*.

Reviewer #1 (Remarks to the Author):

Comments for the manuscript " Revealing the Three-Dimensional Arrangement of Polar Topology in Nanoparticles " by Chaehwa Jeong and colleagues for Nat. Comm.

The significance of topological defects in modern physics is so profound that we are yet to fully grasp all the potential future applications of these enigmatic self-organized local structures. The past two decades have been marked by intense exploration into various types of topological defects in magnetic systems, and more recently, the focus has expanded to include ferroelectric topology, particularly in the study of ferroelectric skyrmions.

The referred paper is devoted the 3D-polarization arrangement using atomic electric tomography (AET) in BTO nanoparticle. Such technique has been also use before to see 3D view of object in many aspects. Here the interesting part is mapping the polarization which is so far has been done by STEM based technique.

The experimental data are quite ok but lack of physics behind that. They just report the observation. It requires quite lot improvement. Here I have some concern about the manuscript.

1. *If they see the vortex, where will the axial component of the polarization.*

Response:

We thank the reviewer for the constructive comment. We agree that a clear description regarding the axial component of the three-dimensional (3D) polarization would be informative for readers. As shown in Revision Fig. 1 and 2, we have presented the slice maps showing the 3D configurations of polarization (in-plane and out-of-plane components) for both Particles 1 and 2. The 3D polarization map clearly reveals the presence of the out-of-plane components of the polarization (i.e., polarization components along the [100] direction) at the vortex cores for both particles, indicating that the vortices are so-called chiral vortices¹.

We have revised Supplementary Fig. 11 of the manuscript accordingly, and Revision Figs. 1 and 2 are now Supplementary Figs. 13 and 14 in our revised manuscript.

Revision Figure 1 | Sliced maps showing the in-plane and out-of-plane polarization configurations for the 8.8 nm BaTiO₃ (Particle 1). **a-c**, 3D Ti atomic displacement maps of representative Ti atomic layers sliced along the [100] (**a**), the [010] (**b**), and the [001] (**c**) directions of the Particle 1. The arrows in the maps indicate the direction of 3D displacement, and their colors reflect the elevation angle between the displacement vector and the plane perpendicular to the [100], [010], and [001] directions for (**a**), (**b**), and (**c**), respectively. A fully red arrow (+90°) points to the [100], [010], and [001] directions for (**a**), (**b**), and (**c**), respectively. A fully blue arrow (−90°) points to the $\bar{1}00$, $0\bar{1}0$, and $00\bar{1}$ directions for (**a**), (**b**), and (**c**), respectively. Note that the vortex core (marked with a black circle) found in (**c**) is the same vortex core (marked with a black circle) observed in (**a**). The distance between the colored arrows is 3.75 Å.

Revision Figure 2 | Sliced maps showing the in-plane and out-of-plane polarization configurations for the 10.1 nm BaTiO₃ (Particle 2). **a-c**, 3D Ti atomic displacement maps of representative Ti atomic layers sliced along the [100] (**a**), the [010] (**b**), and the [001] (**c**) directions of the Particle 2. The arrows in the maps indicate the direction of 3D displacement, and their colors reflect the elevation angle between the displacement vector and the plane perpendicular to the [100], [010], and [001] directions for (**a**), (**b**), and (**c**), respectively. A fully red arrow (+90°) points to the [100], [010], and [001] directions, while a fully blue arrow (-90°) points to the $[\bar{1}00]$, $[0\bar{1}0]$, and $[00\bar{1}]$ directions for (**a**), (**b**), and (**c**), respectively. The distance between the colored arrows is 3.75 Å.

2. *Can they demonstrated also real 3D image of a single vortex and anti-vortex using whole nanoparticle. At the moment it is only 2d slice.*

Response:

We thank the reviewer for the very important comment. As the reviewer suggested, illustrating the 3D images of these polar topologies is indeed crucial, and we believe it plays a significant role in visualizing our research findings. Accordingly, as shown in Revision Fig. 3, we have depicted the 3D polarization distribution across the entire nanoparticle and at each polar topology. Revision Figure 3 has now been included as Supplementary Fig. 12 in our revised manuscript.

Revision Figure 3 | 3D structures of polarization distributions. a-f, 3D polarization configurations of the Particle 1 (a, c) and the Particle 2 (b, d-f). (a, b) show the polarization distributions of the entire nanoparticles. (c) and (d, e) display the zoomed-in views of the vortex core regions of Particle 1 (marked with a blue circle) and Particle 2 (marked with blue and green circles), respectively. (f) displays the zoomed-in view of the antivortex region (marked with a purple circle) found in Particle 2. The orientations of the zoomed-in views are consistent with the orientations of each particle. Note that each polarization vector is represented as a unit vector for visualization purposes.

3. *In particle 1 there are only vortex-like structures while in particle 2 observed vortex and anti-vortex, why? Formation of vortex and antivortex is not clear although theory also tell they verify the experiment.*

Response:

We appreciate the reviewer's comment raising very important points. We fully agree with the reviewer that a more detailed discussion on the size-dependent formation of the vortex and antivortex would enhance the physics for readers. In fact, the formation of vortices within the ferroelectric nanoparticle system arises from the balancing between elastic energy, electrostatic energy, depolarization energy, and gradient energy. These energies vary depending on factors such as boundary conditions, nanoparticle size, and shape, resulting in different non-trivial polar topologies. In particular, for a phase transition in terms of the number of vortices, the size of the nanoparticle is a primary factor, as suggested by multiple simulation works²⁻⁴. Our results are consistent with these simulation predictions in the sense that a smaller particle (Particle 1) shows a single vortex structure, while a larger one (Particle 2) forms multiple vortices. Typically, if two neighboring vortices possess opposite orientations (one clockwise, another one counterclockwise), the polarization vectors in between the vortices are unidirectionally aligned². However, if the neighboring vortices have the same orientation, the direction of the polarization vectors in between the vortices will be frustrated (head-to-head and/or tail-to-tail configurations). In this case, an antivortex is usually formed in between the vortices, stabilizing the energy configuration^{5,6}. These phenomena have been observed in various systems, including ferroelectric^{5,7}, ferromagnetic⁶, and superconducting⁸ systems. As shown in Supplementary Fig. 11b, our results clearly indicate that an antivortex is indeed formed in between two vortices of identical orientation (both are counterclockwise).

This explains the difference in (anti)vortex configuration between the smaller (Particle 1) and larger (Particle 2) nanoparticles.

We have added this discussion into the ‘Discussion and outlook’ section of the manuscript, which now reads “The difference in the number of vortices (one vortex for Particle 1 and two vortices for Particle 2) originates from the size-dependent topological transitions from single vortex-like structures to those of multiple vortices, as predicted from effective Hamiltonian and phase-field simulations^{15,26,49}. In particular, in Particle 2, an antivortex is formed between two neighboring vortices of identical orientation (both are counterclockwise in this case) to create an energetically stable configuration, as observed in other ferroelectric^{50,51}, ferromagnetic⁵², and superconducting⁵³ systems.”.

4. *Author claimed, bigger particle will generate more vortex. Can they take bigger particle and do the same experiment. I am interested to see the vortice pattern over the particle. In fact, this is important for readers as well.*

Response:

We thank the reviewer for the valuable suggestion. Unfortunately, AET is nearly impossible for larger nanoparticles due to issues related to depth of focus (DOF) and dynamic scattering effect. In ADF-STEM-based electron tomography, the entire sample should be in focus when the images at each tilt angle are obtained. Therefore, selecting an appropriate depth of focus based on the sample thickness is important for determining the quality of AET. The depth of focus is inversely proportional to the square of the convergence semi-angle⁹, as described in equation (1) :

$$\text{DOF} = 1.77 \frac{\lambda}{\alpha^2}, \quad (1)$$

here, λ is electron wavelength and α is convergence beam semi-angle. Under our experimental conditions (electron wavelength of 1.97 pm and convergence beam semi-angle of 18.0 mrad), DOF is approximately 10.8 nm, which is appropriate for the AET experiment of nanoparticles with sizes of 8.8 nm and 10.1 nm. However, AET becomes unfeasible for nanoparticles larger than these sizes under the same experimental conditions. Although using a smaller convergence semi-angle can provide a greater DOF, it will also negatively affect the spatial resolution of the ADF-STEM images and the atomic resolution cannot be achieved. Additionally, larger-sized particles exhibit an increased occurrence of multiple scattering that induces non-linear imaging contrast, further deteriorating the resolution. Therefore, performing tomography on nanoparticles larger than 12 nm at atomic resolution is currently unfeasible with the ADF-STEM-based AET implementation. Please note that there are ongoing developments toward overcoming this problem by using the multislice electron tomography (MSET) algorithm¹⁰. In the MSET scheme, measurements can be taken with the sample in an out-of-focus state, eliminating the limitation imposed by DOF on the sample size. Additionally, this algorithm compensates for the nonlinear effects from multiple scattering, making it feasible to observe vortex patterns in larger-sized nanoparticles. We plan to address the polarization distribution of larger nanoparticles in follow-up studies based on the MSET method.

We revised the second paragraph of the ‘Discussion and outlook’ section of our revised manuscript, which now contains the following additional paragraph: “Additionally, a 4D-STEM-based multislice electron tomography algorithm⁶¹ allows measurements to be taken with the sample in the out-of-focus state, overcoming the limitation imposed by the depth of focus on the sample size. Since the multislice-based method can also compensate for the nonlinear effects caused by multiple scatterings, we anticipate that it will allow us to reveal the toroidal orderings in much larger nanoparticles and the transition mechanisms from the vortex states towards the bulk polar states.”.

5. *If I understand correctly, from this method one can not determine the vorticity, chirality and topological number so far. But an object is a topological or not determine by its topological number. They just give some references which I believe is not enough.*

Response:

We thank the reviewer for raising a very important point. This comment has provoked intensive discussions between the authors of this paper, and we believe that our discussions have led to considerable enhancements in the manuscript, thereby increasing its substantial influence on the community engaged in topological polar ordering research. Since we can determine the 3D Ti atomic displacement maps (and the resulting polarization maps), we are certainly capable of calculating physical

quantities, including chirality and topological invariants (e.g., winding number, skyrmion number, and Hopf invariant), based on the polarization maps.

First of all, in our original submission, we already calculated the curl of the in-plane polarization vector throughout the particles and further determined the rotational orientations (clockwise or counterclockwise) of toroidal orderings (Supplementary Fig. 11). We used the term ‘vorticity’ to represent the curl of the in-plane polarization vector, borrowed from fluid dynamics¹¹. However, this definition of vorticity actually differs from the one typically used in skyrmion number calculation¹². To avoid confusion, we decided not to use the word ‘vorticity’ in our manuscript. We have changed the term ‘vorticity’ to ‘curl of in-plane polarization’ in our revised text. Additionally, we used the term ‘winding number’ to represent the quantity used in calculating the skyrmion number (i.e., topological charge) as detailed below.

Second, we quantitatively calculated the local winding numbers by taking a line integral of the change in polarization vector angles over a closed path near different positions in the 2D polarization vector fields (3D vectors in a 2D plane) sliced from our 3D polarization distributions along [100]¹³ (see Methods). We successfully identified the positions of vortices (winding number of +1), convergent hedgehog-type structures (winding number of +1), and antivortices (winding number of -1) throughout the 3D nanoparticles (Revision Fig. 4). We also further verified the chirality of the vortices by combining the previously obtained 2D vortex structures with their local out-of-plane polarization maps (Revision Figs. 1 and 2), and the results were consistent with those calculated from the normalized helicity density map (Revision Fig. 5). Here, normalized helicity density (H_n), commonly used in fluid dynamics¹⁴, is given by

$$H_n = \frac{\mathbf{P} \cdot (\nabla \times \mathbf{P})}{|\mathbf{P}| |\nabla \times \mathbf{P}|}, \quad (2)$$

where \mathbf{P} is the local polarization. The magnitude of the helicity values indicates the chirality of the vortices, and the sign of the normalized helicity density represents the handedness of the identified vortices¹³. Note that the former method (2D vortex structure combined with out-of-plane polarization) depends on the direction of slicing, whereas the latter method (based on helicity density) intrinsically utilizes the full local 3D polarization distribution and does not require slicing. Among the topologically non-trivial structures we identified (non-zero winding numbered objects), the antivortex and hedgehog-type structures are expected to show zero helicities. In case of the vortices, the helicities can be well-defined. For all the vortices appearing in both Particles 1 and 2, the direction of the curl is parallel to the axial component of the polarization (i.e., $H_n > 0$) at the vortex center, indicating that they are all right-handed.

Third, we calculated the Pontryagin charge density (q_x) maps for the 2D polarization vector fields (3D vectors in a 2D plane) sliced from our 3D polarization distributions along [100] direction (Revision Fig. 6). This expression is given by^{12,15}

$$q_x = \frac{1}{8\pi} \epsilon_{ijk} \hat{\mathbf{P}} \cdot (\partial_j \hat{\mathbf{P}} \times \partial_k \hat{\mathbf{P}}), \quad (3)$$

where ϵ_{ijk} is the Levi-Civita symbol, and $\hat{\mathbf{P}}$ is the unit vector representing the direction of local polarizations. Revision Figure 6 shows the presence of clearly finite Pontryagin charge density near the vortex and antivortex areas, indicating that each vortex and antivortex possesses a non-trivial polar topology. To achieve the complete topological classification of 3D polarization vector fields, we further determined the skyrmion numbers for each vortex and antivortex by taking a local 2D surface integral of the Pontryagin charge density^{12,16}, as shown in Revision Fig. 7 (see Methods). In the case of Particle 1, the skyrmion number was found to be approximately +0.3 near the vortex core region. Considering that the corresponding vortex also possesses right-handed chirality, it could be classified as a Bloch-type fractional skyrmion. For Particle 2, as observed in the first representative slice of Revision Fig. 7b, the skyrmion numbers of the two vortices were +0.7 and +0.1, respectively, again characterizing them as Bloch-type fractional skyrmions. The skyrmion number of the antivortex only found in Particle 2 was also identified as -0.4. Additionally, in the case of Particle 2, we confirmed the presence of a polar topology classified as a convergent hedgehog-type fractional skyrmion with a skyrmion number of +0.3 (Revision Fig. 7b). The observed topological charge values deviating from integer or half-integer, which are not allowed in a continuum model, can be attributed to the discrete nature of the atomic structure (which serves as the quanta of polarization distribution in this case) and finite size of the system which limits the integration range. This phenomenon is consistent with experimental measurement of ferromagnetic fractional topological charges along the edge of the magnetic object¹⁷. Furthermore, we observed that the skyrmion number undergoes an abrupt change of its sign (from -0.5 to +0.6) through the region marked with a black square in Revision Fig. 7b. This typically indicates the existence of a Bloch point, where the polarization vanishes. The 3D polarization distribution near the Bloch point is

shown in the Revision Fig.7c, and the skyrmion number for the surrounding area is +1, indicating that the Bloch point structure has a diverging spiral configuration.

Finally, since we have obtained the full 3D polarization distributions, the Hopf invariant, representing the integer topological charge for a hopfion (3D counterparts of skyrmions), can be calculated using the following equation¹³ :

$$N_H = - \int \mathbf{F} \cdot \mathbf{A} dV. (4)$$

Here, $\mathbf{F}_i = \frac{1}{8\pi} \epsilon_{ijk} \hat{\mathbf{P}} \cdot (\partial_j \hat{\mathbf{P}} \times \partial_k \hat{\mathbf{P}})$ and \mathbf{A} is defined as a gauge field satisfying $\mathbf{F} = \nabla \times \mathbf{A}$. The Hopf invariants for Particle 1 and Particle 2 were calculated to be -0.002 and -0.011 , respectively, indicating that neither of the particles is likely to possess a hopfion.

In overall, for the first time, we were able to observe topologically non-trivial 3D polar structures at the atomic scale and quantitatively calculate their topological invariants to classify them into different categories including fractional skyrmions and a spiral Bloch point structures.

We added this discussion in the ‘Discussion and outlook’ part of the manuscript, which now reads “The normalized helicity density (H_n) maps also clearly show non-zero helicity density values near the vortex centers, confirming their chirality (Supplementary Fig. 15). The helicity density map further allows us to determine the handedness of each chiral vortex by examining its sign⁵⁴. Positive helicity density can be clearly observed for all the vortices appearing in both Particles, indicating that all the vortices are right-handed, which is also evident from the fact that the direction of the curl is parallel to the axial component of the polarization (Supplementary Figs. 12-15). The antivortex and hedgehog-type structures are expected to show zero helicities in ideal configurations⁵⁴, consistent with our experimental findings (Supplementary Fig.15). Although the existence of vortices and antivortices, along with their chirality/handedness, can be qualitatively analyzed based on the curl of in-plane polarization and helicity density maps, these methods cannot quantitatively provide polar topological information of measured 3D polarization distributions. Therefore, for quantitative analysis of topological structures in our 3D polarization distributions, we further calculated topological invariants (e.g., winding number, skyrmion number, and Hopf invariant) with the circle-equivalent and sphere-equivalent order-parameter spaces⁵⁴ (Methods). First, we calculated the topological invariants defined for 2D polarization vector fields (2D vectors in a 2D plane). From the 2D polarization vector fields sliced from our 3D polarization distributions along [100] direction, local winding numbers were calculated by taking a line integral of the change in the arguments of the in-plane polarization vectors over a closed path⁵⁴. As can be seen in Fig. 5a, b, we successfully identified the positions of vortices (winding number of +1), convergent hedgehog-type structures (winding number of +1), and antivortices (winding number of -1), along with their 3D distributions, which are consistent with the qualitative findings described above. Second, we continued to calculate the topological invariants defined for 3D vectors in a 2D plane, starting from the Pontryagin charge density maps, again for the sliced polarization distributions along the [100] direction (Methods). Figure 5c, d shows the presence of clearly finite Pontryagin charge density near the vortex and antivortex areas, indicating that each vortex and antivortex possesses a non-trivial polar topology. To achieve the complete topological classification, we further determined the skyrmion numbers for each vortex and antivortex by taking a local 2D surface integral of the Pontryagin charge density^{55,56}, as shown in Supplementary Fig. 16. In the case of Particle 1, the skyrmion number was found to be approximately +0.3 near the vortex core region. Considering that the corresponding vortex also possesses right-handed chirality, it could be classified as a Bloch-type fractional skyrmion. For Particle 2, as observed in the first representative slice of Supplementary Fig. 16b, the skyrmion numbers of the two vortices were +0.7 and +0.1, respectively, again characterizing them as Bloch-type fractional skyrmions. The skyrmion number of the antivortex, only found in Particle 2, was also identified as -0.4. Additionally, in the case of Particle 2, we confirmed the presence of a polar topology classified as a convergent hedgehog-type fractional skyrmion with a skyrmion number of +0.3 (Supplementary Fig. 16b). The observed topological charge values deviating from integer or half-integer, which are not allowed in a continuum model, can be attributed to the discrete nature of the atomic structure (which serves as the quanta of polarization distribution in this case) and finite size of the system which limits the integration range. This phenomenon is consistent with the experimentally measured emergence of ferromagnetic fractional topological charges along the edge of the magnetic object⁵⁷. Furthermore, we observed that the skyrmion number undergoes an abrupt sign change (from -0.5 to +0.6) through the region marked with black squares in Supplementary Fig. 16b. This typically indicates the existence of a Bloch point, where the polarization vanishes. The 3D polarization distribution near the Bloch point is shown in the Supplementary Fig. 16c, and the skyrmion number for the surrounding area is +1, indicating that the Bloch point structure has a diverging spiral configuration. Finally, since we have obtained the full 3D polarization distributions, the Hopf invariant⁵⁴, indicative of the integer topological charge of a hopfion (3D counterparts of skyrmions),

can be calculated. Particles 1 and 2 showed the Hopf invariant of -0.002 and -0.011 , respectively, indicating that neither of the particles is likely to possess a hopfion. Through these analyses, for the first time, we were able to observe topologically non-trivial 3D polar structures at the atomic scale and quantitatively calculate their topological invariants to classify them into different categories including fractional skyrmions and spiral Bloch point structures. Note that the topological analyses described above require 2D slicing of the 3D polarization distributions (except for the Hopf invariant), and the result can potentially depend on the choice of slicing direction. Therefore, our demonstration of full 3D polarization mapping necessitates a novel approach that can classify the topological structures within 3D polar nanostructures more robustly.

We also included ‘Analysis of swirling characteristics of the polarization vector fields’ part and ‘Topological structures analysis’ part in the Methods section, which now read “Chirality can be determined through the normalized helicity density (H_n), a mathematical concept adapted from fluid dynamics, which is defined as¹⁸

$$H_n = \frac{\mathbf{P} \cdot (\nabla \times \mathbf{P})}{|\mathbf{P}| |\nabla \times \mathbf{P}|}, \quad (5)$$

where \mathbf{P} is local polarization.” and “The characteristics of topological structures in 3D systems can be described by topological invariants (e.g., winding number, skyrmion number, and Hopf invariant). Winding number is defined as a topological invariant in the circle-equivalent order-parameter space, and is used to distinguish between vortices (winding number of $+1$) and antivortices (winding number of -1) in a 2D system. The winding map was calculated by taking a line integral of the change in the arguments of the in-plane polarization vectors over a closed path (square loop with a side length of 3.75 \AA) near different positions in the 2D polarization vector fields (2D vectors in a 2D plane) sliced from our 3D polarization distributions along $[100]$ direction⁸⁸.

Skyrmion number (N_{sk}) is defined as a topological invariant in the sphere-equivalent order-parameter space and is used to characterize the swirling structure of a skyrmion. To obtain skyrmion number, we first calculated the Pontryagin charge density (q_x) maps for the 2D polarization vector fields (3D vectors in a 2D plane) sliced from our 3D polarization distributions along the $[100]$ direction (Fig. 5c,d). This expression is given by^{55,56}

$$q_x = \frac{1}{8\pi} \epsilon_{ijk} \hat{\mathbf{P}} \cdot (\partial_j \hat{\mathbf{P}} \times \partial_k \hat{\mathbf{P}}) \Big|_{j=y, k=z} = \frac{1}{4\pi} \hat{\mathbf{P}} \cdot (\partial_y \hat{\mathbf{P}} \times \partial_z \hat{\mathbf{P}}), \quad (6)$$

where ϵ_{ijk} is the Levi-Civita symbol, and $\hat{\mathbf{P}}$ is the unit vector representing the direction of local polarizations. For the topological characterization of 3D polarization vector fields, we further calculated the skyrmion numbers for each vortex and antivortex, which are given by

$$N_{sk} = \int q_x dA_x = \int \frac{1}{4\pi} \hat{\mathbf{P}} \cdot (\partial_y \hat{\mathbf{P}} \times \partial_z \hat{\mathbf{P}}) dydz, \quad (7)$$

The skyrmion map was determined by performing a local 2D surface integral of the Pontryagin charge density over a circular area with a radius of 10 \AA for Particle 1 and 13.75 \AA for Particle 2, centered around each position in the (100) plane (Supplementary Fig. 16). Note that skyrmion numbers at each vortex and antivortex can vary depending on the chosen integration areas. Therefore, we calculated multiple skyrmion numbers at the locations of vortices, antivortices, and hedgehog-type structures for several different choices of the integration area and checked their trends. These trends clearly indicate that the skyrmion numbers saturate beyond certain integration areas, which were chosen as the area for our analysis given above. To calculate the skyrmion number at the Bloch point, the left-hand side of equation (7) can be generalized into 3D based on the divergence theorem¹⁹.

$$N_{sk} = \int q_i dA_i = \int \frac{3}{4\pi} \partial_x \hat{\mathbf{P}} \cdot (\partial_y \hat{\mathbf{P}} \times \partial_z \hat{\mathbf{P}}) dx dy dz. \quad (8)$$

Hopf invariant (N_H), representing the integer topological charge for a hopfion (3D counterparts of skyrmions), can be calculated using the following equation^{54,90}.

$$N_H = - \int \mathbf{F} \cdot \mathbf{A} dV. \quad (9)$$

Here, $\mathbf{F}_i = \frac{1}{8\pi} \epsilon_{ijk} \hat{\mathbf{P}} \cdot (\partial_j \hat{\mathbf{P}} \times \partial_k \hat{\mathbf{P}})$ and \mathbf{A} is defined as a gauge field satisfying $\mathbf{F} = \nabla \times \mathbf{A}$. The vector potential \mathbf{A} was determined in momentum space using the Coulomb gauge condition ($\nabla \cdot \mathbf{A} = 0$), and based on this, the Hopf invariant was calculated in momentum space⁹⁰. ”, respectively. Revision Figures 4 and 6 have now been included as Fig. 5, and Revision Figs. 5 and 7 have been included as Supplementary Fig. 15, and 16 in our revised manuscript, respectively.

Revision Figure 4 | Representative 2D slices through the nanoparticles showing local winding numbers. **a, b,** Winding number maps of representative Ti atomic layers along the [100] direction of Particle 1 (8.8 nm) (**a**), and Particle 2 (10.1 nm) (**b**). The in-plane directions of the Ti displacements are indicated by black arrows. The red, yellow, and blue colors represent the vortex (winding number of +1), hedgehog-type structures (winding number of +1), and antivortex (winding number of -1), respectively. Note that the winding number itself cannot distinguish between stripe domains, flux-closure domains, hedgehog-type, and vortex (or antivortex) structures; the identification of vortex structure requires more information regarding the local polarization distributions. Accordingly, we marked in green the areas where the winding number is +1 or -1, but which are neither vortices, hedgehog-type structures, nor antivortices. The distance between the arrows is 3.75 Å.

Revision Figure 5 | Representative 2D slices through the nanoparticles showing normalized helicity density. **a, b,** Normalized helicity density maps of representative Ti atomic layers along the [100] direction of Particle 1 (8.8 nm) (**a**), and Particle 2 (10.1 nm) (**b**). The in-plane directions of the Ti displacements are overlaid (black arrows). Note that the red (larger than 0) and blue (smaller than 0) colors represent right-handed and left-handed chirality, respectively. The regions of vortices, antivortices, and hedgehog-type structures are marked with red, blue, and yellow circles, respectively. The distance between the arrows is 3.75 Å.

Revision Figure 6 | Representative 2D slices through the nanoparticles showing Pontryagin charge density distribution. **a, b**, Pontryagin charge density maps of representative Ti atomic layers along the [100] direction of Particle 1 (8.8 nm) (**a**), and Particle 2 (10.1 nm) (**b**). The in-plane directions of the Ti displacements are overlaid (black arrows). Vortices, antivortices, and hedgehog-type structures are marked with red, blue, and yellow circles, respectively. The distance between the arrows is 3.75 Å.

Revision Figure 7 | Representative 2D slices through the nanoparticles showing skyrmion numbers. **a, b**, Skyrmion number maps of representative Ti atomic layers along the [100] direction of Particle 1 (8.8 nm) (**a**), and Particle 2 (10.1 nm) (**b**). The in-plane directions of the Ti displacements are overlaid (black arrows). Vortices, antivortices, and hedgehog-type structures are marked with red, blue, and yellow circles, respectively. The distance between the arrows is 3.75 Å. The region where the sign of skyrmion number rapidly changes (from -0.5 to 0.6) has been marked with black squares. **c**, The 3D polarization configuration of the volume corresponding to $1.25 \times 1.25 \times 1.25 \text{ \AA}^3$ around the Bloch point (marked with a blue sphere) within the region marked with black squares in (**b**).

6. They claimed counterclockwise vortex exists in consecutive layers of the bottom half of the nanoparticle. But haven't discussed why only in the bottom half, what about another part of the particle that does not get a vortex-like structure?

Response:

We greatly appreciate this insightful comment. The reviewer has indeed highlighted a point that we had not fully considered. The formation of a ferroelectric vortex state typically occurs under poor screening conditions, while good screening conditions lead to the formation of a polar state. Real ferroelectric samples possess various sources of screening, which include charge carriers arising from the presence of oxygen vacancies. These can partially screen the depolarization field even under open-circuit boundary conditions, hindering the formation of a complete vortex structure. In these ambiguous screening conditions, it is theoretically anticipated that the polar order and toroidal order can coexist, leading to a state called polar-toroidal multiorder (PTMO)²⁰. In the PTMO state, the vortex center becomes off-centered rather than being at the center of the particle. This creates toroidal order around the vortex center and polar ordering away from it (see the Revision Fig. 8 below).

Revision Figure 8 | A dipole configuration in PTMO state. The coexistence of a vortex state with an off-centered vortex center (marked with a blue arrow) and a polar state (marked with a red arrow) can be clearly observed. This figure is adapted from Ji, Y. *et al.* Phys. Rev. B 100, 014101 (2019).

Similarly, as can be seen in Fig. 2a, our results show that the top half of the nanoparticle exhibits a polar order, while the bottom half exhibits a vortex order. This observation directly corresponds to the coexistence of these two states induced by the off-centering of the vortex core towards the bottom part. Furthermore, for Particle 1, since the vortex core near the center slice of the nanoparticle is also off-centered towards the surface, and the direction of the toroidal moment is not perfectly aligned along [100] direction, the vortex core can only penetrate the bottom half of the nanoparticle. Such off-centering of the vortex core and mixed vortex and polar ordering can also be observed in Particle 2, as illustrated in Fig. 3a.

We revised the third paragraph of ‘3D local polarization maps of the nanoparticles’ section of our revised manuscript, which now reads “The in-plane displacement direction maps clearly reveal that a unidirectional polar order dominates in the top half of the particle while a counterclockwise vortex exists in the bottom half. This corresponds to the theoretically predicted polar-toroidal multiorder (PTMO) state, in which the polar and toroidal orders coexist³⁶. In real samples, due to the presence of various screening sources such as charge carriers arising from oxygen vacancies, the electrostatic boundary condition can exhibit a partially screened state, rather than being perfectly short-circuited or fully open-circuited. In this case, the vortex core can be off-centered rather than being at the center of the particle, resulting in toroidal order near the vortex center and polar ordering away from it.”.

7. Why not other particles of the same size range selected, do results reproduce in other nanoparticles or only in these 2 nanoparticles which are reported?

Response:

We thank the reviewer for the suggestion. In this study, we aimed to experimentally observe not only the distribution of polarization within ferroelectric nanoparticles but also the size-dependent topological transitions from a single vortex to multiple vortices. Therefore, we selected nanoparticles of two different sizes for tomography. However, due to the relatively weaker bonding nature of BaTiO₃ nanoparticles compared to metallic nanoparticles for which most AET works have been performed so far, the experiments have been extremely challenging and time-consuming. Previous AET experiments were mainly conducted on metallic nanoparticles with an electron dose of at least $7 \times 10^5 \text{ e } \text{\AA}^{-2}$ (see refs²¹⁻²³). Since the kinetic energies that can be transferred to the oxygen atoms in BaTiO₃ nanoparticles are 12

times larger than those in typical metallic nanoparticles²⁴, electrons of 300 keV energy can more easily damage BaTiO₃ nanoparticles compared to metallic nanoparticles. Therefore, it was necessary to reduce the electron dose by about seven times compared to previous AET experiments. Due to these dose constraints, it took us three years to obtain the two tomograms at atomic resolution. Consequently, due to time constraints, it is very challenging to conduct additional experiments to verify reproducibility within the three-month timeframe given to us for the revision. Although we are unable to conduct additional experiments, we verified the robustness of the observed vortex under electron beam exposure with additional analysis as follows. According to previous studies²⁵, the cation substitution in BaTiO₃ nanoclusters remains unchanged up to an electron dose of $8.78 \times 10^5 e \text{ \AA}^{-2}$. Since we used an electron dose that is ten times lower than this, we anticipate that no significant structural changes would occur. Additionally, to assess the effects of surface diffusion during tilt series measurements, we conducted simulations of tomography experiments by applying random perturbation to surface atoms. As can be seen in Supplementary Fig. 3j-l, consistent internal vortex structures can be obtained through tomographic reconstruction even considering the effects of surface diffusion. The consistent formation of vortices at the same locations and numerous STEM-based experiments and simulations^{2,3,26} suggest that the vortex structure is not an artifact of an electron beam but is an intrinsic topological polar structure of the actual BaTiO₃ nanoparticles.

8. *Does the vortex-like structure remain stable over time or changes*

Response:

We appreciate the reviewer bringing up important points. The discussion on the stability of the vortex-like structure in our study is equivalent to considering the effects of the electron beam on ferroelectric materials, because we did not alter temperature or electric field environments during the experiments. The first effect of the electron beam is knock-on damage, where atom displacement may result from the kinetic energy transferred from an electron beam to an atom during a collision²⁷. As mentioned in response to the previous comment (response to comment #7), the knock-on damage, under the electron dose we used, does not alter the displacement of cation atoms. We checked this by measuring the zero-degree ADF-STEM images three times (before, in the middle of, and after the tilt series measurement), and no visible structural changes were observed especially for the core regions of the nanoparticles (Supplementary Fig. 3a-f). Moreover, even after tomography simulations that considered actual atomic diffusion, the vortex structure appears, suggesting that the effects of knock-on damage are likely to be averaged out during the reconstruction process (Supplementary Fig. 3j-l). The second effect is domain switching induced by the electron beam, which arises from the electric fields generated by charge accumulation. It is known from a previous study that a minimum electron dose of $1.3 \times 10^6 e \text{ \AA}^{-2}$ is required for electron beam-induced domain formation for free-standing 50 nm BaTiO₃ nanoparticles, and a minimum dose of $5.3 \times 10^7 e \text{ \AA}^{-2}$ is needed for domain switching²⁸. In our study, we used an electron dose that is at least 10 times smaller than what is required for electron beam-induced domain formation or switching. Therefore, the impact of this factor is also not a significant concern. Consequently, we can infer that the vortex-like structure remains stable over time.

We have included this discussion in the ‘STEM data acquisition’ part of the Methods section, which now reads “Furthermore, the domain formation or switching induced by electric fields resulting from charge accumulation due to the electron beam is also not a significant concern in this study, as it requires at least ten times larger electron dose⁵⁹.”.

9. *Also, author could update their conclusion, means, if they do such evaluatory measurements on ferroelectric skyrmion domain, what will be the exciting.*

Response:

We thank the reviewer for the constructive suggestion. As recommended, we have added more detailed discussions about intriguing issues related to ferroelectric topological domains in the last paragraph of ‘Discussion and outlook’ part of our revised manuscript, which now reads “Moreover, due to the unique topological nature, polar skyrmions are expected to show enhanced stability over time, and their distinct topological protection renders them highly metastable in environments with field polarization with minimal energy dissipation⁶³. If combined with the PTMO state we observed, our observation of non-trivial polar structures (fractional skyrmions and a spiral Bloch point structure) in 3D ferroelectric nanostructures without epitaxial strain can open a door towards even higher capacity storage devices, since it allows doubling of the memory capacity by separately controlling the linear polarity and chirality, which can carry a bit of information. We hope that our approaches will provide a more comprehensive

understanding towards the nature of 3D polar orderings in strain-free nanostructures, and open a pathway leading to functional devices along with further development of switching and reading schemes.”.

Reviewer #2 (Remarks to the Author):

Identifying three-dimensional atomic morphology of topological domains and the distribution of the ferroelectric polarization in real space is of scientific and technological importance. In this manuscript, Chaehwa Jeong et. al. applied atomic electron tomography and unveiled the three-dimensional arrangement of ferroelectric polarizations in BaTiO₃ nanoparticles. They show that the polarization distribution within BaTiO₃ nanoparticles exhibits a vortex state, and the distribution of vortex is correlated with the size of the nanoparticles showing a transition from the single state to the multiple states. Although the authors showed preliminary structures of the three-dimensional vortices, there are still some limitations at the present stage. I suggest the authors address the following concerns before the work is considered for publication in Nature Communications.

1. *The authors showed an average atomic displacement of Ti within the nanoparticles to be 18.8 pm. This value is much larger than the bulk value (~ 5pm), and such a significant displacement would result in a substantial ferroelectric polarization (~100 μC cm⁻²), which is conflict with the estimated value in the manuscript.*

Response:

We thank the reviewer for pointing out an important issue. Firstly, as the reviewer pointed out, the average Ti atomic displacements were 18.8 pm for Particle 1 and 29.0 pm for Particle 2, significantly greater than the bulk value. Nonetheless, in nanostructured BaTiO₃ systems of reduced size, an increase in Ti atomic displacement has been experimentally confirmed and they are consistent with our values^{25,29,30}. For example, a mean Ba-Ti displacement of ~30 pm was found from a 4 × 4 nm² BaTiO₃ nanocluster²⁵. Second, we have carefully reviewed our calculations for converting the Ti atomic displacements into ferroelectric polarizations. Since we could only discern the atomic positions of cations, the positions of oxygen atoms were inferred using simultaneous equations (1) and (2) below (they are also provided in the Methods section)³⁰⁻³².

$$P_s = \kappa(\delta_{Ti} - \delta_O). \quad (1)$$

$$P_s = \frac{e}{V} (2Z_{O\perp}^* \delta_O + Z_{O\parallel}^* \delta_O + Z_{Ti}^* \delta_{Ti}). \quad (2)$$

From this, we could determine the linear scale factor (α) between the polarization and Ti atomic displacement as shown in equation (3).

$$P_s = \alpha \delta_{Ti}. \quad (3)$$

We found that the scale factor (α) of 1.89 we previously used was underestimated due to lattice constant mismatch. The Born effective charges we previously used for calculating the scale factor were obtained based on a cubic phase BaTiO₃ with a lattice constant of 3.9 Å³³. However, the BaTiO₃ used in our study showed, as evidenced by powder X-ray diffraction (PXRD) analysis, a cubic phase with a lattice constant of 4.0 Å. Therefore, during the revision, we employed the Born effective charges ($Z_{O\perp}^* = -2.13$, $Z_{O\parallel}^* = -5.75$, and $Z_{Ti}^* = 7.29$) calculated for a cubic structure with the lattice constant of 4.00 Å. Consequently, the scale factor (α) was determined as 2.12 ± 0.69 , which is approximately 12% larger than the previous value. Based on this scale factor and the bulk Ti atomic displacement (9 pm)³⁴, we can estimate the bulk polarization as a value of 0.19 ± 0.06 C m⁻², consistent with the bulk polarization value 0.25 C m⁻² within the error bar. Using the new scale factor, we updated the net polarizations reported in our manuscript to the new values of 0.40 ± 0.13 C m⁻² for Particle 1 and 0.61 ± 0.20 C m⁻² for Particle 2. Accordingly, the magnitudes of the toroidal moments were also updated as 0.22 ± 0.07 e Å⁻¹ for Particle 1 and 0.51 ± 0.16 e Å⁻¹ for Particle 2.

2. *The ferroelectric polarization is strong correlated with the lattice in perovskite oxide ferroelectrics. The 18.8 pm displacement corresponds to a considerable c/a value in the BTO unit cell. However, the authors claimed that the unit cell of BTO is close to cubic.*

Response:

We thank the reviewer for raising very important points. As the reviewer pointed out, in a ferroelectric BaTiO₃ crystal at room temperature, the Ti atomic displacement along the direction of the unit cell's tetragonal distortion typically induces spontaneous polarization. Therefore, there is a strong correlation between tetragonality and atomic displacement in bulk systems.

Firstly, we reverified whether the BaTiO₃ used in this study had tetragonality through PXRD and error analysis. As shown in the inset of Revision Fig. 9, diffraction peak splitting typically observed from tetragonally distorted systems³⁵ was not observed in the two-theta angular range corresponding to the (002) diffraction peak. Considering that the FWHM of the (002) peak is 0.607°, tetragonal splitting

should have been observed if the two-theta angle difference between the two split peaks is greater than 0.607° (i.e., c/a of 1.013), as can be seen in Revision Fig. 10a. We then further checked the minimum c/a ratio required for peak splitting. Assuming tetragonal distortion (i.e., the PXRD peak arising from the a -axis is twice as large as that from the c -axis), tetragonal splitting can only be observed when the two theta angles between two peaks are greater than 0.303 (c/a being 1.006) under the given FWHM, and below this c/a ratio, the peak splitting cannot be detected (Revision Fig. 10b). Therefore, our PXRD suggests that our BaTiO₃ nanoparticles exhibit a long-range cubic phase (i.e., no (002) peak splitting and c/a of unity), but the precision of our (c/a) value is approximately 0.006 as described above, and there is a possibility that our particles have tetragonal distortion within this c/a ratio range.

Since we have full 3D cation atomic structures of the nanoparticles, we can perform local analysis regarding the tetragonality, rather than relying on the ensemble-averaged PXRD results. Therefore, we further calculated the tetragonality of the local (short-range) structure based on local tetragonal lattice fitting (Methods).

As can be seen in Revision Fig. 11, there are localized areas with high tetragonality that cannot arise from a cubic phase. Our findings, where the reduced size of BaTiO₃ exhibits distortion in the short-range but simultaneously shows a cubic phase in the long-range, are consistent with the results in previous studies³⁵. Additionally, when comparing the tetragonality map with the 3D displacement magnitude map, it is evident that areas with weak tetragonality (i.e., c/a ratio being close to 1) can also possess large Ti displacement and resulting polarization. In nanostructured ferroelectric systems, unlike in bulk materials, an unusually strong atomic displacement is observed in STEM studies, and it does not always correlate with the tetragonality^{31,36}. Therefore, the commonly known correlation between tetragonality and atomic displacement cannot fully explain the ferroelectric behavior in nanostructured ferroelectric systems.

We have included this discussion in the fifth paragraph of ‘3D local polarization maps of the nanoparticles’ part of our revised manuscript and ‘Local tetragonal lattice fitting’ part in the Methods section, which now reads “Since the nanoparticles show substantial cation displacement and consequent local polarization which typically accompany local tetragonal distortion, we further performed local structural analysis regarding the local tetragonality (Methods). Although no long-range tetragonal distortion can be observed from the PXRD analysis (see Supplementary Fig. 1g and Methods), there are localized areas with high tetragonality that cannot arise from a cubic phase (Supplementary Fig. 9). Our findings, where the reduced size of BaTiO₃ exhibits distortion in the short-range but simultaneously shows a cubic phase in the long-range, are consistent with the results in previous studies⁴⁰. Additionally, a comparison of the tetragonality map (Supplementary Fig. 9) with the 3D displacement magnitude map (Figs. 2 and 3) reveals that areas with low tetragonality (i.e., c/a ratio being close to 1) can also exhibit large Ti atomic displacement and resulting polarization. In nanostructured ferroelectric systems, unlike in bulk materials, an unusually strong atomic displacement has been observed in previous STEM studies, and it does not always correlate with the tetragonality, consistent with our findings^{41,42}.”, and “For the analysis of local tetragonality (c/a ratio), we performed a local tetragonal lattice fitting using the following procedures. First, for each Ti atom, we identified eight nearest-neighbor Ba atoms based on the globally fitted bcc lattice configuration (see ‘Assignment of experimental 3D atomic positions to ideal bcc lattices’ section of Methods). Second, we performed three tetragonal lattice vector fittings to the atoms assigned to the lattice, for [100], [010], and [001] as potential c -axis directions, respectively. This fitting involved parameters for translation, 3D rotation, and scaling, and was done to minimize the error between the measured atomic positions (which includes only the 8 nearest neighbor atoms for each Ti atom) and the corresponding lattice sites of the fitted tetragonal crystal lattice. Among these, the c -vector direction that minimized the errors between the measured atomic positions and the corresponding lattice sites of the fitted tetragonal crystal lattice was determined as the c -axis for that particular local structure. This procedure was applied to all Ti atoms which have eight nearest neighbor atoms. As a result, the average c/a ratio for each nanoparticle was determined to be 1.002 ± 0.002 for Particle 1 and 1.002 ± 0.001 for Particle 2. Note that the standard error propagation, based on the precision of the atomic coordinates, was used to determine the error bars in this process.”, respectively. And the Revision Figs. 9 and 11 are now Supplementary Figs. 1g and 9 in our revised manuscript.

For the analysis of local tetragonality (c/a ratio), we performed a local tetragonal lattice fitting using the following procedures. First, for each Ti atom, we identified eight nearest-neighbor Ba atoms based on the globally fitted bcc lattice configuration (see “Assignment of experimental 3D atomic positions to ideal bcc lattices” section of Methods). Second, we performed three tetragonal lattice vector fittings to the atoms assigned to the lattice, for [100], [010], and [001] as potential c -axis directions, respectively. This fitting involved parameters for translation, 3D rotation, and scaling, and was done to minimize the error between the measured atomic positions (which includes only the 8 nearest neighbor atoms for each

Ti atom) and the corresponding lattice sites of the fitted tetragonal crystal lattice. Among these, the c -vector direction that minimized the errors between the measured atomic positions and the corresponding lattice sites of the fitted tetragonal crystal lattice was determined as the c -axis for that particular local structure. This procedure was applied to all Ti atoms which have eight nearest neighbor atoms. As a result, the average c/a ratio for each nanoparticle was determined to be 1.002 ± 0.002 for Particle 1 and 1.002 ± 0.001 for Particle 2. Note that the standard error propagation, based on the precision of the atomic coordinates, was used to determine the error bars in this process.

Revision Figure 9 | Powder X-ray diffraction pattern of the BaTiO_3 nanoparticles. The inset shows the X-ray diffraction pattern in the two-theta angular range of 44° to 46° corresponding to the (002) diffraction peak. The black dots represent the measured PXRD data and the blue solid lines indicate the measured data after Gaussian smoothing.

Revision Figure 10 | The measured (002) diffraction peak of the BaTiO_3 nanoparticles overlaid with simulated Gaussian functions. **a, b,** Powder X-ray diffraction pattern of the BaTiO_3 nanoparticles around (002) peaks overlaid with simulated Gaussian functions with the tetragonal splitting of 0.607° (**a**) and 0.303° (**b**) in two-theta. The black dots represent the measured PXRD data, the blue and red solid lines indicate the simulated Gaussian functions along c - and a -axes, and the green solid lines represent the summation of the blue and red lines.

Revision Figure 11 | Representative sliced maps showing the tetragonality map. a, b, Tetragonality (c/a) maps of representative Ti atomic layers sliced along the [100] direction of Particle 1 (8.8 nm) (a), and Particle 2 (10.1 nm) (b). The in-plane directions of the Ti displacements are overlaid (black arrows). Note that the blue and red colors represent the local c/a ratio. The precisions of the local c/a ratios are 0.012 for Particle 1 and 0.011 for Particle 2, determined through standard error propagation. Scale bar, 2 nm.

3. *The precision of the atomic coordinates is estimated to be 38.3 and 34.4 pm, which are much larger than ideal displacement of Ti columns in BTO, indicating that such methodology may be not suitable to characterize the polarization behavior in this material. Ferroelectric with larger spontaneous polarization might be much more suitable for investigating the 3D polarization distribution.*

Response:

We thank the reviewer for pointing out this very critical issue. This is a very important issue, and we tried to address this point in our original manuscript, but it seems that the explanation was not sufficient enough. We acknowledge that the estimated precision of 38.3 pm for Particle 1 and 34.4 pm for Particle 2 is indeed much larger than the typical Ti atomic displacement of BaTiO₃ (9 pm). This might suggest limitations in accurately characterizing the polarization behavior of the material at the single-atom level. To overcome this, we applied local averaging of the displacements for several atoms based on the Gaussian kernel. In this way, we were able to substantially enhance precision since averaging (i.e., more sampling) can reduce the impact of random noise or fluctuations in the data. However, as a trade-off, the spatial resolution is reduced to approximately 0.9 nm compared to the atomic scale (sub-Angstrom). After applying kernel-based local averaging, the precisions of atomic displacements were improved to 4.9 pm for Particle 1 and 4.3 pm for Particle 2 and this level of precision is sufficient to describe the global polarization distribution and vortex ordering. Additionally, this can be confirmed in the atomic displacement maps after kernel averaging obtained from tomographic reconstruction with Multislice simulation results. As can be seen in Supplementary Fig. 3j-l, consistent internal vortex structures are attainable in multislice simulation results, even with the atomic coordinates having root-mean-square deviations (RMSDs) greater than 30 pm. This demonstrates that, despite the RMSDs of 30 pm, applying kernel-based local averaging allows us to sufficiently reproduce their polarization behavior.

We updated the manuscript accordingly, and the corresponding part of the manuscript now reads “Note that the application of kernel-based local averaging effectively improves precision, reducing the error in the locally-averaged displacement field to about 4.9 pm for Particle 1 and 4.3 pm for Particle 2. This level of precision is adequate to describe the global polar ordering behavior, considering the typical ferroelectric local displacement level (tens of picometers). However, as a trade-off, it results in a slightly lower 3D spatial resolution of approximately 0.9 nm, which is not at the atomic scale³⁰.”.

4. *The authors analyzed the ferroelectric vortex structures in BTO nanoparticles layer by layer, and the results are still remain in two-dimensional planes, without any additional 3D information of the lattice ferroelectric polarization.*

Response:

We thank the reviewer for very constructive suggestions. We fully agree that the 3D information on polarization would be informative for the readers of our paper. As mentioned in the response to Reviewer 1’s comments #1 and #2, we have represented the 3D polarization configuration, illustrating both in-plane and out-of-plane polarization directions, as shown in Revision Figs. 1 and 2, and we have also depicted the 3D polarization direction in the entire nanoparticle as well as at each topological defect (vortex and antivortex core) in Revision Fig. 3. These figures are now included in our revised manuscript as Supplementary Figs. 12, 13, and 14.

References

1. Wang, Y. J., Tang, Y. L., Zhu, Y. L. & Ma, X. L. Entangled polarizations in ferroelectrics: A focused review of polar topologies. *Acta Materialia* **243**, 118485 (2023).
2. Naumov, I. I., Bellaiche, L. & Fu, H. Unusual phase transitions in ferroelectric nanodisks and nanorods. **432**, 4 (2004).
3. Mangeri, J. *et al.* Topological phase transformations and intrinsic size effects in ferroelectric nanoparticles. *Nanoscale* **9**, 1616–1624 (2017).
4. Pitike, K. C. *et al.* Metastable vortex-like polarization textures in ferroelectric nanoparticles of different shapes and sizes. *Journal of Applied Physics* **124**, 064104 (2018).
5. Abid, A. Y. *et al.* Creating polar antivortex in PbTiO₃/SrTiO₃ superlattice. *Nat Commun* **12**, 2054 (2021).
6. Ruotolo, A. *et al.* Phase-locking of magnetic vortices mediated by antivortices. *Nature Nanotech* **4**, 528–532 (2009).
7. Louis, L., Kornev, I., Geneste, G., Dkhil, B. & Bellaiche, L. Novel complex phenomena in ferroelectric nanocomposites. *J. Phys.: Condens. Matter* **24**, 402201 (2012).
8. Chibotaru, L. F., Ceulemans, A., Bruyndoncx, V. & Moshchalkov, V. V. Symmetry-induced formation of antivortices in mesoscopic superconductors. *Nature* **408**, 833–835 (2000).
9. Kiguchi, T., Yamaguchi, Y., Tashiro, S., Sato, K. & Konno, T. J. Effect of Focal Depth of HAADF-STEM Imaging on the Solute Enriched Layers in Mg Alloys. *Mater. Trans.* **56**, 1633–1638 (2015).
10. Lee, J., Lee, M., Park, Y., Ophus, C. & Yang, Y. Multislice Electron Tomography Using Four-Dimensional Scanning Transmission Electron Microscopy. *Phys. Rev. Applied* **19**, 054062 (2023).
11. Lugt, H. J. & Gollub, J. P. *Vortex Flow in Nature and Technology*. *American Journal of Physics* **53**, (1985).
12. Nagaosa, N. & Tokura, Y. Topological properties and dynamics of magnetic skyrmions. *Nature Nanotech* **8**, 899–911 (2013).
13. Junquera, J. *et al.* Topological phases in polar oxide nanostructures. *Rev. Mod. Phys.* **95**, 025001 (2023).
14. Tang, Y. & Liu, Y. VR helicity density and its application in turbomachinery tip leakage flows. *Chinese Journal of Aeronautics* **35**, 1–17 (2022).
15. Hierro-Rodriguez, A. *et al.* Revealing 3D magnetization of thin films with soft X-ray tomography: magnetic singularities and topological charges. *Nat Commun* **11**, 6382 (2020).
16. Nahas, Y. *et al.* Discovery of stable skyrmionic state in ferroelectric nanocomposites. *Nat Commun* **6**, 8542 (2015).

17. Jena, J. *et al.* Observation of fractional spin textures in a Heusler material. *Nat Commun* **13**, 2348 (2022).
18. Levy, Y., Degani, D. & Seginer, A. Graphical visualization of vortical flows by means of helicity. *AIAA Journal* **28**, 1347–1352 (1990).
19. Rana, A. *et al.* Three-dimensional topological magnetic monopoles and their interactions in a ferromagnetic meta-lattice. *Nat. Nanotechnol.* **18**, 227–232 (2023).
20. Ji, Y., Chen, W. J. & Zheng, Y. Crossover of polar and toroidal orders in ferroelectric nanodots with a morphotropic phase boundary and nonvolatile polar-vortex transformations. *Phys. Rev. B* **100**, 014101 (2019).
21. Yang, Y. *et al.* Deciphering chemical order/disorder and material properties at the single-atom level. *Nature* **542**, 75–79 (2017).
22. Zhou, J. *et al.* Observing crystal nucleation in four dimensions using atomic electron tomography. *Nature* **570**, 500–503 (2019).
23. Yang, Y. *et al.* Atomic-scale identification of the active sites of nanocatalysts. Preprint at <https://doi.org/10.48550/arXiv.2202.09460> (2022).
24. Egerton, R. F., Li, P. & Malac, M. Radiation damage in the TEM and SEM. *Micron* **35**, 399–409 (2004).
25. Bencan, A. *et al.* Atomic scale symmetry and polar nanoclusters in the paraelectric phase of ferroelectric materials. *Nat Commun* **12**, 3509 (2021).
26. Fu, H. & Bellaiche, L. Ferroelectricity in Barium Titanate Quantum Dots and Wires. *Phys. Rev. Lett.* **91**, 257601 (2003).
27. Jiang, N. Electron irradiation effects in transmission electron microscopy: Random displacements and collective migrations. *Micron* **171**, 103482 (2023).
28. Barzilay, M. & Ivry, Y. Formation and manipulation of domain walls with 2 nm domain periodicity in BaTiO₃ without contact electrodes. *Nanoscale* **12**, 11136–11142 (2020).
29. Sato, Y. *et al.* Atomic-Scale Observation of Titanium-Ion Shifts in Barium Titanate Nanoparticles: Implications for Ferroelectric Applications. *ACS Appl. Nano Mater.* **2**, 5761–5768 (2019).
30. Abrahams, S. C., Kurtz, S. K. & Jamieson, P. B. Atomic Displacement Relationship to Curie Temperature and Spontaneous Polarization in Displacive Ferroelectrics. *Phys. Rev.* **172**, 551–553 (1968).
31. Jia, C.-L. *et al.* Unit-cell scale mapping of ferroelectricity and tetragonality in epitaxial ultrathin ferroelectric films. *Nature Mater* **6**, 64–69 (2007).

32. Zhong, W., King-Smith, R. D. & Vanderbilt, D. Giant LO-TO splittings in perovskite ferroelectrics. *Phys. Rev. Lett.* **72**, 3618–3621 (1994).
33. Ghosez, Ph., Gonze, X., Lambin, Ph. & Michenaud, J.-P. Born effective charges of barium titanate: Band-by-band decomposition and sensitivity to structural features. *Phys. Rev. B* **51**, 6765–6768 (1995).
34. Kwei, G. H., Lawson, A. C., Billinge, S. J. L. & Cheong, S. W. Structures of the ferroelectric phases of barium titanate. *J. Phys. Chem.* **97**, 2368–2377 (1993).
35. Shi, C. *et al.* Barium titanate nanoparticles: Short-range lattice distortions with long-range cubic order. *Phys. Rev. B* **98**, 085421 (2018).
36. Kim, S.-D. *et al.* Inverse size-dependence of piezoelectricity in single BaTiO₃ nanoparticles. *Nano Energy* **58**, 78–84 (2019).

REVIEWER COMMENTS

Reviewer #1 (Remarks to the Author):

I have carefully reviewed the comments on review points and modifications in the manuscript titled "Revealing the Three-Dimensional Arrangement of Polar Topology in Nanoparticles." This manuscript has garnered significant attention due to its detailed explanation of the 3D imaging of polar topology. The exploration of the polar topology in nanoparticle system with atomic electron tomography demonstrates the robustness of these topological defects, making them promising for applications where stability is critical. The pathways to tune such defects, like creation, manipulation, and annihilation, will guide the exploration of more ferroelectric systems, such as thin films. The authors have already well articulated strategies to make the application based by independent of epitaxial strain, offering potential for practical applications utilizing contact-free switchable toroidal moments. Several future aspects are also presented to overcome challenges, such as the real-space experimental observation of dipole texture and achieving higher density topological defects for increased information density at room temperature without any external stimuli. After all, the key observations are nicely presented in terms of theoretical points, and systematic 3D construction of topological defects are well explained.

In this manuscript, my review points are well explained. The reported results are robust and well-understood. The manuscript is now interesting and clear, catering to the esteemed community of the journal.

Reviewer #2 (Remarks to the Author):

In the revised manuscript, the authors added additional analysis and tried to confirm the topological ordering in the BTO nanoparticles. These analyses clarified some of my previous confusion, but some other confusions still remain:

1. The significant ion displacement observed in the nanoparticles can be attributed to the size effect. However, it is intriguing that the second nanoparticle, despite having a larger size, exhibits an apparently larger ion displacement.
2. The authors have stated that the BTO nanoparticles exhibit a long-range cubic phase based on PXRD experiments, while the local lattice fitting process suggests a short-range tetragonal phase. According to the tetragonal splitting analysis in the PXRD results, the c/a ratio should be less than 1.006. However, in the revised Figure 11, the average local c/a ratio appears to be over 1.02, despite the authors' claim of an averaged c/a ratio of only $1.002 \pm 0.002 (\pm 0.001)$ in the revised manuscript. This discrepancy between the long-range and short-range results raises the question of why they conflict. It may be helpful to include an additional histogram of the c/a ratio to provide further insights.
3. In the revised Figure 11, the polarization vectors are superimposed on the c/a maps to illustrate the relationship between local c/a values and local polarizations. However, it is confusing to observe that the direction of polarization is not correlated with the tetragonality. As depicted in the figure, in areas with the same c/a ratio (red, $c/a > 1$), the polarization direction shows no discernible regularity, randomly pointing along the [001] or [010] direction. In previously reported ferroelectric topological domains with tetragonal-like structures, the polarization direction consistently aligns with the orientation of the c -axis. Therefore, the inconsistent results between the polarization and c/a parameters undermine the reliability of the overall 3D distributions of the polar vectors in the BTO nanoparticles.
4. The authors enhanced the precision of the atomic coordinates through kernel-based local averaging. In the original polar maps without any averaging methods, the polarization distribution appears to have little regular pattern. However, after applying a Gaussian kernel, the polarization distribution exhibits a vortex state. Additionally, the averaged R-factors were 0.22 and 0.17, indicating a significant discrepancy between the experimental tilt series data and the reconstructed volume data. Therefore, it raises the question of whether the vortex state mentioned in the manuscript corresponds to the actual presence of vortex topological structures in the sample.
5. The authors only demonstrated the direction of the polarization near the center of the vortex state, without providing any information on magnitude of the polarization.
6. In Figure 2 and Figure 3, it appears that the PTMO state is only present on a few atomic layers.

However, the reported vortex structures exhibit an elongated tubular shape. Therefore, the term "vortex domain" may not accurately describe this particular structure. A thorough analysis of the polar characteristics, particularly in the vicinity of, including the PTMO state, is necessary to provide a comprehensive 3D representation of the polar behavior. This will enable an accurate determination of the type of polar topology.

RESPONSE TO REVIEWERS' COMMENTS

First, we thank the reviewers for taking the time and effort to provide constructive comments, which have been very helpful in making improvements to the manuscript.

Please find below our response to the reviewers' comments on our manuscript "*Revealing the Three-Dimensional Arrangement of Polar Topology in Nanoparticles*". We have carefully addressed all the reviewers' criticisms and suggestions and trust that with the presented changes, the manuscript is now acceptable for publication in *Nature Communications*.

Reviewer #1 (Remarks to the Author):

I have carefully reviewed the comments on review points and modifications in the manuscript titled "Revealing the Three-Dimensional Arrangement of Polar Topology in Nanoparticles." This manuscript has garnered significant attention due to its detailed explanation of the 3D imaging of polar topology. The exploration of the polar topology in nanoparticle system with atomic electron tomography demonstrates the robustness of these topological defects, making them promising for applications where stability is critical. The pathways to tune such defects, like creation, manipulation, and annihilation, will guide the exploration of more ferroelectric systems, such as thin films. The authors have already well articulated strategies to make the application based by independent of epitaxial strain, offering potential for practical applications utilizing contact-free switchable toroidal moments. Several future aspects are also presented to overcome challenges, such as the real-space experimental observation of dipole texture and achieving higher density topological defects for increased information density at room temperature without any external stimuli. After all, the key observations are nicely presented in terms of theoretical points, and systematic 3D construction of topological defects are well explained. In this manuscript, my review points are well explained. The reported results are robust and well-understood. The manuscript is now interesting and clear, catering to the esteemed community of the journal.

We are deeply grateful for the reviewer's valuable suggestions, which have significantly contributed to improvements in various aspects of our work.

Reviewer #2 (Remarks to the Author):

In the revised manuscript, the authors added additional analysis and tried to confirm the topological ordering in the BTO nanoparticles. These analyses clarified some of my previous confusion, but some other confusions still remain:

- The significant ion displacement observed in the nanoparticles can be attributed to the size effect. However, it is intriguing that the second nanoparticle, despite having a larger size, exhibits an apparently larger ion displacement.*

Response:

We appreciate the reviewer highlighting a crucial point of our study. In our previous manuscript, we compared the magnitude of the averaged atomic displacement vector, i.e., calculating the average before determining its magnitude ($|\langle \delta_{Ti} \rangle|$, where $\langle \delta_{Ti} \rangle$ represents the average of atomic displacement vector, and ‘|’ signifies the magnitude), to assess the overall atomic displacement. However, we realized that comparing $|\langle \delta_{Ti} \rangle|$ of the two different nanoparticles can be misleading for evaluating atomic displacements, as they can cancel out depending on the topological distribution of the atomic displacements. Therefore, we calculated the magnitudes of the local displacement vectors and obtained their distributions to estimate the overall displacement behaviors (Revision Fig. 1); this method is more consistent with previous studies on nanosized ferroelectrics^{1,2}. The magnitude of the overall Ti atomic displacement was determined by fitting our distributions to a generalized extreme value distribution^{3,4}, considering the asymmetry of the measured Ti atomic displacements, resulting in 43.5 pm for Particle 1 and 40.5 pm for Particle 2, respectively. Consistent with previous size effect studies, we observed a larger Ti atomic displacement in the smaller particle (Particle 1). Furthermore, this significant Ti atomic displacement aligns with previous studies: a 35 ± 18 pm average Ti displacement in 4×4 nm² BaTiO₃ nanocluster⁵ and a spontaneous polarization of $100 \mu\text{C cm}^{-2}$, which corresponds to a displacement of 47 pm according to our equation (3), observed in a 9 nm BaTiO₃ nanoparticle⁶.

We have included this discussion in the fourth and sixth paragraphs of the ‘3D local polarization maps of the nanoparticles’ section of our revised manuscript, which now read “We also calculated the overall magnitude of the displacement vector by fitting our distribution to a generalized extreme value distribution⁴¹, considering the asymmetry of the measured Ti atomic displacements (Supplementary Fig. 7), which was determined to be 43.5 pm.”, and “The overall magnitude of the displacement vector for Particle 2 was determined to be 40.5 pm (Supplementary Fig. 7), slightly smaller than that for Particle 1, consistent with previous studies of the size effect on the magnitude of the displacement vector⁴².”. Revision Figure 1 has now been included as Supplementary Fig. 7 in our revised manuscript.

Revision Figure 1 | Distribution of the magnitude of the kernel-averaged local Ti atomic displacements. a, b, Histograms of the magnitude of the local Ti atomic displacements for Particle 1 (**a**), and Particle 2 (**b**), respectively. The red solid line represents the fitted generalized extreme value distribution, with the location parameter (μ) of the fitted distribution being 43.5 pm for Particle 1 and 40.5 pm for Particle 2, respectively. Note that the displacement vector at each atom position is interpolated from the kernel-averaged displacement vector map.

2. The authors have stated that the BTO nanoparticles exhibit a long-range cubic phase based on PXRD experiments, while the local lattice fitting process suggests a short-range tetragonal phase. According to the tetragonal splitting analysis in the PXRD results, the c/a ratio should be less than 1.006. However, in the revised Figure 11, the average local c/a ratio appears to be over 1.02, despite the authors' claim of an averaged c/a ratio of only $1.002 \pm 0.002 (\pm 0.001)$ in the revised manuscript. This discrepancy between the long-range and short-range results raises the question of why they conflict. It may be helpful to include an additional histogram of the c/a ratio to provide further insights.

Response:

We thank the reviewer for pointing out an important issue. Please note that PXRD can only measure an ensemble-averaged c/a value of all the unit cells in the system. Our technique can measure the c/a ratios of individual unit cells, albeit with slightly lower precision (larger error bar), from which we can obtain the full distribution of the c/a values of unit cells in our particles (Revision Fig. 2). The reviewer expressed concern regarding the discrepancy between the long-range, ensemble-averaged c/a value and the local c/a values. However, these values do not necessarily need to be consistent. Ensemble-averaged PXRD, giving a c/a ratio less than 1.006, does not necessarily mean that all the individual unit cells in the system should have a c/a ratio less than 1.006. Depending on the broadness of the distribution, local values can be very different from the average. Therefore, we believe that we need to compare the averaged c/a ratio with that obtained from PXRD to check the consistency between our AET result and PXRD result, rather than directly comparing local AET values to the ensemble-averaged PXRD value. Regarding the error bars of our averaged c/a ratio, although the individual c/a values exhibit a broad distribution (some local unit cells have c/a values beyond 1.02, as the reviewer pointed out; see Revision Fig. 2), the averaged c/a value can be calculated with much higher precision because we have the c/a information from numerous individual unit cells (i.e., statistical sampling points) to be averaged, which will lead to higher precision inversely proportional to the square root of the number of unit cells⁷.

We have revised this discussion in the last paragraph of the '3D local polarization maps of the nanoparticles' section of our revised manuscript, which now reads "Since the nanoparticles show substantial cation displacement and consequent local polarization which typically accompany local tetragonal distortion, we further performed local structural analysis regarding the local tetragonality (Methods). Although no distinct long-range tetragonal distortion can be observed from the PXRD analysis (see Supplementary Fig. 1g and Methods), there are localized areas with high tetragonality that cannot arise from a cubic phase (Supplementary Figs. 10 and 11). Our findings, where the nanosized BaTiO_3 exhibits a cubic-like feature when ensemble averaged but shows substantial distortion on the short-range scale, are consistent with the results in previous studies⁴³." Revision Figure 2 is now Supplementary Fig. 11 in our revised manuscript.

Revision Figure 2 | Distribution of the kernel averaged c/a ratios. **a, b**, Histograms of the individual c/a ratios for Particle 1 (**a**), and Particle 2 (**b**), respectively. Note that the c/a ratio at each atom position is interpolated from the kernel-averaged tetragonality (c/a) map.

3. *In the revised Figure 11, the polarization vectors are superimposed on the c/a maps to illustrate the relationship between local c/a values and local polarizations. However, it is confusing to observe that the direction of polarization is not correlated with the tetragonality. As depicted in the figure, in areas with the same c/a ratio (red, c/a > 1), the polarization direction shows no discernible regularity, randomly pointing along the [001] or [010] direction. In previously reported ferroelectric topological domains with tetragonal-like structures, the polarization direction consistently aligns with the orientation of the c-axis. Therefore, the inconsistent results between the polarization and c/a parameters undermine the reliability of the overall 3D distributions of the polar vectors in the BTO nanoparticles.*

Response:

We are grateful for the reviewer's insightful comments. As far as we understand, the reviewer's point (polarization being aligned along the *c*-axis) is correct only for so-called "c-phase" of nanosized ferroelectrics. Numerous studies on ferroelectric nanostructures with different phases (v-phase, h-phase, etc.) have reported that the polarization direction does not always align with the orientation of the *c*-axis^{8–10}, particularly in non-trivial topological structures like the vortex phase.

We have revised this discussion in the last paragraph of the '3D local polarization maps of the nanoparticles' section of our revised manuscript, which now reads "In nanostructured ferroelectric systems, unlike in bulk materials, an unusually strong atomic displacement has been observed in previous studies, and it does not always correlate with the tetragonality or *c*-axis orientations, consistent with our findings^{44–46}."

4. *The authors enhanced the precision of the atomic coordinates through kernel-based local averaging. In the original polar maps without any averaging methods, the polarization distribution appears to have little regular pattern. However, after applying a Gaussian kernel, the polarization distribution exhibits a vortex state. Additionally, the averaged R-factors were 0.22 and 0.17, indicating a significant discrepancy between the experimental tilt series data and the reconstructed volume data. Therefore, it raises the question of whether the vortex state mentioned in the manuscript corresponds to the actual presence of vortex topological structures in the sample.*

Response:

We appreciate the reviewer highlighting such a critical aspect of reliability. Note that the polarization distribution before kernel averaging (Supplementary Figs. 6 and 8) already shows the vortex structures at the locations consistent with those observed after kernel averaging (Figs. 2 and 3). To further verify whether the polarization distribution exhibits the same vortex state before and after Gaussian kernel averaging, we calculated the winding number, as illustrated in Revision Fig. 3. These results indicate the consistent presence of substantial, non-zero winding numbers in regions with vortices even before kernel averaging. Regarding the R factor, it should be considered that we did not perform the usual R-factor minimization in our structural analysis^{11,12}, and only cation atoms were used for the forward-projection calculations (excluding oxygen atoms). Even in this situation, we achieved R-factors between 0.20 and 0.25, which are generally considered acceptable in the fields of crystallography and AET^{13–16}, indicating the reliability of our atomic structures.

We have included this discussion in the 'Ti atomic displacement field and polarization field calculation' part of Methods section, which now reads "Note that we applied kernel averaging to atomic displacement vectors and polarization to increase the precision of the individual displacement vectors. Regardless of kernel averaging, the polarization distribution exhibits substantial vortex structures (Supplementary Figs. 6 and 8). To further confirm whether the polarization distribution maintains the same vortex state prior to Gaussian kernel averaging, we calculated the winding number both before and after kernel averaging, as shown in Supplementary Fig. 19. These results indicate the consistent presence of substantial, non-zero winding numbers in vortex regions, even before the application of kernel averaging.". Revision Figure 3 is now Supplementary Fig. 19 in our revised manuscript.

Revision Figure 2 | Representative 2D slices through the nanoparticles showing winding numbers. **a, b,** Winding number maps of representative Ti atomic layers along the [100] direction of Particle 1 (8.8 nm), before **(a)** and after **(b)** applying a Gaussian kernel. **c, d,** Winding number maps of representative Ti atomic layers along the [100] direction of Particle 2 (10.1 nm), before **(c)** and after **(d)** applying a Gaussian kernel. The areas with high winding number values were marked with red circles. Note that in **(b)** and **(d)**, the displacement vector at each atom position is interpolated from the kernel-averaged displacement vector map.

5. *The authors only demonstrated the direction of the polarization near the center of the vortex state, without providing any information on magnitude of the polarization.*

Response:

We thank the reviewer for the comments. The reviewer's comment about the magnitude of the polarization is addressed in Figs. 2b and 3b. Here, we confirmed that the magnitudes of 3D local polarizations (i.e., 3D Ti atomic displacements) are consistent with the previously reported spontaneous polarization of $100 \mu\text{C cm}^{-2}$ for 9 nm BaTiO_3 nanoparticles⁶ and the polarization values are appropriately suppressed near the vortex cores, as expected¹⁷.

The corresponding discussion in the third paragraph of the '3D local polarization maps of the nanoparticles' section of our revised manuscript has been revised as "The range of 3D Ti displacements (and the resulting polarization values) is consistent with the previously observed values in the nanosized BaTiO_3 ^{38,39}. The region near the core of the vortex (indicated by blue arrows) shows relatively smaller local polarizations compared to other regions where the vortex does not appear, which is also in agreement with the result from phase-field simulations²⁷."

6. *In Figure 2 and Figure 3, it appears that the PTMO state is only present on a few atomic layers. However, the reported vortex structures exhibit an elongated tubular shape. Therefore, the term "vortex domain" may not accurately describe this particular structure. A thorough analysis of the polar characteristics, particularly in the vicinity of, including the PTMO state, is necessary to provide a comprehensive 3D representation of the polar behavior. This will enable an accurate determination of the type of polar topology.*

Response:

We appreciate the reviewer for pointing out very important aspects of our research. As the reviewer pointed out, our full 3D determination of local polarization distribution can provide a comprehensive 3D representation of the polar behavior. Therefore, we conducted thorough analyses of our entire 3D distributions using all known polar characterization methods, which include local 3D polarization mapping, global net polarization calculation, toroidal moment calculation, curl of in-plane polarization analysis for local vortices and anti-vortices, normalized helicity density mapping, 2D winding number calculation (topological invariants defined for 2D polarization vector fields [2D vectors in a 2D plane]), skyrmion number calculation (topological invariants defined for 3D polarization vector fields [3D vectors in a 2D plane]), and 3D topological analyses of Hopf invariant and Bloch point. These analyses were conducted for the entire 3D polarization distributions of the nanoparticles in this study, which of course includes the regions in the vicinity of the PTMO state. Our layer-by-layer 3D mappings (Figs. 2 and 3) clearly show that our vortex structures indeed exhibit an elongated tubular shape, but they are off-centered, resulting in the PTMO state with the tubular vortex line penetrating only a few atomic layers. Also, as the reviewer suggested, we are not using the term "vortex domain" in our manuscript. As far as we know, this level of 3D polar topological analysis has never been achieved before, and we believe that these results can provide new insights to the researchers in this field.

References

1. Polking, M. J. *et al.* Ferroelectric order in individual nanometre-scale crystals. *Nat. Mater.* **11**, 700–709 (2012).
2. Sato, Y. *et al.* Atomic-Scale Observation of Titanium-Ion Shifts in Barium Titanate Nanoparticles: Implications for Ferroelectric Applications. *ACS Appl. Nano Mater.* **2**, 5761–5768 (2019).
3. Singh, V. P. Generalized Extreme Value Distribution. in *Entropy-Based Parameter Estimation in Hydrology* (ed. Singh, V. P.) 169–183 (Springer Netherlands, Dordrecht, 1998). doi:10.1007/978-94-017-1431-0_11.
4. Lu, X. *et al.* Visualizing Magnetic Order in Self-Assembly of Superparamagnetic Nanoparticles. Preprint at <http://arxiv.org/abs/2401.01284> (2024).
5. Bencan, A. *et al.* Atomic scale symmetry and polar nanoclusters in the paraelectric phase of ferroelectric materials. *Nat. Commun.* **12**, 3509 (2021).
6. Basun, S. A., Cook, G., Reshetnyak, V. Yu., Glushchenko, A. V. & Evans, D. R. Dipole moment and spontaneous polarization of ferroelectric nanoparticles in a nonpolar fluid suspension. *Phys. Rev. B* **84**, 024105 (2011).
7. Huang, B., Bates, M. & Zhuang, X. Super-Resolution Fluorescence Microscopy. *Annu. Rev. Biochem.* **78**, 993–1016 (2009).
8. Naumov, I. & Fu, H. Vortex-to-Polarization Phase Transformation Path in Ferroelectric Pb(ZrTi)O₃ Nanoparticles. *Phys. Rev. Lett.* **98**, 077603 (2007).
9. Kim, S.-D. *et al.* Inverse size-dependence of piezoelectricity in single BaTiO₃ nanoparticles. *Nano Energy* **58**, 78–84 (2019).
10. Pavlenko, M. A. *et al.* Phase Diagram of a Strained Ferroelectric Nanowire. *Crystals* **12**, 453 (2022).
11. Yang, Y. *et al.* Deciphering chemical order/disorder and material properties at the single-atom level. *Nature* **542**, 75–79 (2017).
12. Zhou, J. *et al.* Observing crystal nucleation in four dimensions using atomic electron tomography. *Nature* **570**, 500–503 (2019).
13. Joosten, R. P., Chinae, G., Kleywegt, G. J. & Vriend, G. 3.23 - Protein Three-Dimensional Structure Validation. in *Comprehensive Medicinal Chemistry II* (eds. Taylor, J. B. & Triggle, D. J.) 507–530 (Elsevier, Oxford, 2007). doi:10.1016/B0-08-045044-X/00096-1.
14. Kregel, U. & Imberty, A. Chapter 2 - Crystallography and Lectin Structure Database. in *Lectins* (ed. Nilsson, C. L.) 15–50 (Elsevier Science B.V., Amsterdam, 2007). doi:10.1016/B978-044453077-6/50003-X.
15. Maveyraud, L. & Mourey, L. Protein X-ray Crystallography and Drug Discovery. *Molecules* **25**, 1030 (2020).
16. Lee, J., Jeong, C. & Yang, Y. Single-atom level determination of 3-dimensional surface atomic structure via neural network-assisted atomic electron tomography. *Nat. Commun.* **12**, 1962 (2021).
17. Mangeri, J. *et al.* Topological phase transformations and intrinsic size effects in ferroelectric nanoparticles. *Nanoscale* **9**, 1616–1624 (2017).

REVIEWERS' COMMENTS

Reviewer #2 (Remarks to the Author):

I find the authors have addressed all my concerns, therefore I am happy to recommend this work for publication in Nature Communications.

RESPONSE TO REVIEWERS' COMMENTS

First, we thank the reviewer for taking the time and effort to provide constructive comments, which have been very helpful in making improvements to the manuscript.

Please find below our response to the reviewer's comment on our manuscript "*Revealing the Three-Dimensional Arrangement of Polar Topology in Nanoparticles*". We have carefully addressed the reviewer's criticisms and suggestions and trust that with the presented changes, the manuscript is now acceptable for publication in *Nature Communications*.

Reviewer #2 (Remarks to the Author):

I find the authors have addressed all my concerns, therefore I am happy to recommend this work for publication in Nature Communications.

Response:

We appreciate the reviewer's constructive comments, which has greatly improved our manuscript.